# Hippocampal replay of experience at real-world speeds

**Eric L Denovellis[1,2,3], Anna K Gillespie[2,3], Michael E Coulter[2,3], Marielena Sosa[4], Jason E Chung[5], Uri T Eden[6], Loren M Frank[1,2,3]\***

[1]Howard Hughes Medical Institute, University of California, San Francisco, San Francisco, United States; [2]Departments of Physiology and Psychiatry, University of California, San Francisco, San Francisco, United States; [3]Kavli Institute for Fundamental Neuroscience, University of California, San Francisco, San Francisco, United States; [4]Department of Neurobiology, Stanford University School of Medicine, Stanford, United States; [5]Department of Neurological Surgery, University of California, San Francisco, San Francisco, United States; [6]Department of Mathematics and Statistics, Boston University, Boston, United States

**Abstract** Representations related to past experiences play a critical role in memory and decision-making processes. The rat hippocampus expresses these types of representations during sharp-wave ripple (SWR) events, and previous work identified a minority of SWRs that contain 'replay' of spatial trajectories at ~20x the movement speed of the animal. Efforts to understand replay typically make multiple assumptions about which events to examine and what sorts of representations constitute replay. We therefore lack a clear understanding of both the prevalence and the range of representational dynamics associated with replay. Here, we develop a state space model that uses a combination of movement dynamics of different speeds to capture the spatial content and time evolution of replay during SWRs. Using this model, we find that the large majority of replay events contain spatially coherent, interpretable content. Furthermore, many events progress at real-world, rather than accelerated, movement speeds, consistent with actual experiences.

\*For correspondence: loren@phy.ucsf.edu

**Competing interests:** The authors declare that no competing interests exist.

## Introduction

The brain has the remarkable ability to store and retrieve representations of past events, enabling memories of the past to influence future behavior. These memory processes depend critically on the hippocampus, where rapid plasticity during an experience is thought to drive the initial encoding of representations of events (*Eichenbaum and Cohen, 2004*). Subsequently, hippocampal 'replay' of stored representations during both slow-wave sleep and periods of immobility during waking is thought to contribute to the longer term storage and updating of event memories in distributed hippocampal-neocortical circuits (*Frankland and Bontempi, 2005*; *Carr et al., 2011*; *Joo and Frank, 2018*).

The canonical example of 'replay' is seen in the rodent, where hippocampal cells preferentially fire at specific locations in an environment, and thus ensembles of cells fire in sequence as the animal moves through a series of locations. When the animal is asleep or immobile, hippocampal cells can be reactivated during a 'sharp-wave ripple' (SWR) event. A subset of SWRs contain sequential firing similar to that seen during a previous experience, and thus are thought to 'replay' these previous experiences. Importantly, previous work has reported that these sequential firing events proceed at an average speed of ~10 meters per second, about 20x faster than the animal's usual movement speed (*Nádasdy et al., 1999*; *Lee and Wilson, 2002*; *Davidson et al., 2009*; *Karlsson and Frank, 2009*).

While the existence of these sequential events is well established, the current consensus is that only a minority (~5–45%) of hippocampal SWRs contain statistically identifiable, sequential replay (*Davidson et al., 2009*; *Michon et al., 2019*; *Shin et al., 2019*; *Kaefer et al., 2020*; *Tingley and Peyrache, 2020b*). One might therefore conclude that all memories that are stored or updated by SWRs correspond to sequences of locations. Our subjective experience, however, suggests that we can retrieve memories of individual locations without having to mentally traverse a long distance. Such memories would seem to be useful, in that they could encode the stimuli and values associated with a given place irrespective of the path used to get there. If the rodent hippocampus is capable of storing those sorts of memories then one might expect to see SWRs where the neural activity corresponds not to a rapid trajectory through space, but instead to either a stable pattern associated with a single location or perhaps a brief pattern more similar to one that occurs during a real experience.

Interestingly, there is evidence for SWRs where the spiking corresponds to a single location. Specifically, some SWRs contain spiking of neurons associated with single locations where animals are immobile (*Yu et al., 2017*), although two reports suggested that events that represent a single location are only seen in young animals (*Farooq and Dragoi, 2019*; *Muessig et al., 2019*). Other studies have reported activity that, when combined across an entire event, seems to correspond roughly to a single location, although the dynamics of these events is typically not investigated (*Dupret et al., 2010*). Thus, while it would seem useful to replay memories associated with single locations or a small region of the environment, the prevalence of that type of replay in awake animals remains unclear.

Our uncertainty stems in part from the dominant approaches used to identify the content of replay events. These approaches typically involve multiple steps and assumptions about the nature of replay, which are most commonly characterized using the standard 'Bayesian' decoder. First, an encoding model is constructed based on spiking activity during movement, most often using only spikes that have been clustered as single units (putative single neurons). Then, a subset of SWRs or events with high activity levels are selected based on a threshold for event size chosen by the experimenter (*Foster and Wilson, 2006*; *Diba and Buzsáki, 2007*; *Karlsson and Frank, 2009*; *Stella et al., 2019*). A decoding algorithm is then applied to the spikes within these events, yielding a set of position probability distributions for each time bin. Current approaches use either overlapping or non-overlapping time bins whose size is also chosen by the experimenter. Finally, the most commonly used approaches for detecting sequential replay involve fitting a line to the resulting set of probability distributions, which relies on the assumption that the representations progress at a constant speed (*Foster and Wilson, 2006*; *Diba and Buzsáki, 2007*; *Davidson et al., 2009*; *Karlsson and Frank, 2009*; *Carr et al., 2012*; *Ólafsdóttir et al., 2017*; *Tang et al., 2017*; *Drieu et al., 2018*; *Shin et al., 2019*; *Tingley and Buzsáki, 2020a*; *Bhattarai et al., 2020*). A statistical test is then used to determine whether the line fit to the data is better than a line fit to shuffled versions of the data, where the shuffled version of the data represent an implicit definition of a 'random' sequence.

While the standard approach identifies constant speed events, it does not consider events that are rejected by the statistical test. This is problematic because it has the potential to mischaracterize real events that do not move at constant speeds, such as those that change direction or are discontinuous (*Liu et al., 2018*). Furthermore, the use of large, fixed-size temporal bins acts as a boxcar smoother that limits the potential movement speeds of the representation. For example, with 20 ms time bins and 3 cm position bins, an event can only move in 1.5 m/s increments (one or more bins in 20 ms) between time steps. The linear fit is also highly dependent on the estimation of event boundaries, such as the start and end times of the SWR, because the fit depends on all the data over the course of the event. Approaches that focus on the order of cell activity within each event (*Lee and Wilson, 2002*; *Gupta et al., 2010*) allow for a relaxation of that linear assumption, but replace it with a loss of statistical power due to either ignoring all but the first spike from each cell or using an arbitrarily smoothed version of the spike train. These approaches also do not provide information about the dynamics of the underlying spatial representation. Moreover, these approaches often exclude events that have stationary representations of a single location (*Yu et al., 2017*; *Farooq and Dragoi, 2019*).

Recognizing the problems with the linear fit, several studies have moved away from the constant velocity assumption, using the most probable location at each time bin (*Pfeiffer and Foster, 2013*;

*Wu and Foster, 2014*; *Grosmark and Buzsáki, 2016*; *Carey et al., 2019*; *Kaefer et al., 2020*). For example, using this approach, *Pfeiffer and Foster, 2015* found awake replay events containing alternation between representations of a single location and sequential spatial trajectories. On the other hand, *Stella et al., 2019* reported that replays during sleep are spatially continuous and follow Brownian diffusion dynamics. However, both methods still used large time bins and neither took into account the uncertainty of the decoded estimates, making it hard to identify the source of the different conclusions.

An ideal approach to identifying and quantifying the dynamics of replay would circumvent these problems. It would use all the available spiking data to yield the most accurate decoded positions. It would provide a moment-by-moment estimate of position and dynamics that would not be dependent on the estimation of SWR start and end times and could rapidly adjust to changes in dynamics. It would use very small temporal bins (1 or 2 ms) to allow for very rapid representational movement and would provide information about the certainty of the decoded estimates. It would be able to capture a range of movement dynamics including stationary or unchanging representations, trajectories that progress through space at constant or variable speeds, and disorganized, spatially incoherent events. It would provide a robust statistical assessment of confidence for each dynamic. Finally, where assumptions are made, it would provide well-defined parameters whose values could be explored systematically to understand their influence on the results.

We developed a state space model that achieves all those goals. State space models are a well-understood, well-known statistical solution to the problems described above. By mathematically modeling the relationship between the data and latent dynamics, state space models make the assumptions of the model explicit and interpretable. Our model goes beyond previous approaches (*Deng et al., 2016*; *Maboudi et al., 2018*) by characterizing the spatial representations during SWRs as a mixture of three underlying patterns of movement dynamics: stationary trajectories, continuous trajectories that can progress at many times the typical speed of the animal, and spatially fragmented trajectories. We show how this model can take advantage of clusterless decoding—which relates multiunit spike waveform features to position without spike sorting—giving us more information about the population spiking activity. We apply this model to spiking data during SWR events from 10 rats, enabling a direct comparison to previous work.

We find that the large majority of SWRs contain spatially coherent content; that is, trajectories that are spatially concentrated at each moment in time and have no large discontinuities in position. Surprisingly, while the expected high-speed, sequential replay events were identified, the most common category of events expressed representations that moved at slower speeds, more consistent with real-world experiences. These findings illustrate the power of state space models and provide a new understanding of the nature of hippocampal replay.

## Results

### Overview of the model

We begin with an application of the state space model to simulated data, both to validate the model and to provide intuition (*Figure 1*). We simulate 19 Poisson spiking cells with Gaussian place fields on a 180 cm virtual linear track. Each place field has a 36 cm variance and a 15 Hz peak firing rate, and fields are spaced every 10 cm along the virtual track. We then construct a 280 ms spiking sequence (*Figure 1A*) and apply our model to the sequence. For the first 60 ms of this sequence, a single place cell fires repeatedly, resulting in the extended representation of a single location. For the second 190 ms of the sequence, the cells fire in sequential spatial order, representing a fast moving trajectory across the virtual linear track. For the last 30 ms of the sequence, the cells fire in an incoherent spatial order. These three firing patterns represent three different types of movement dynamics that could be expressed during SWRs, which we call stationary, continuous, and fragmented, respectively. The goal of our model is to characterize SWRs in terms of a mixture of these three dynamics at every time point.

Decoding the spiking sequence requires specifying two elements: the data model—how the spikes relate to position—and the movement dynamic models—how the position can change over time in each movement dynamic. For the data model, our decoder is the same as the standard ('Bayesian') decoder (*Davidson et al., 2009*; *Pfeiffer and Foster, 2015*; *Stella et al., 2019*). We

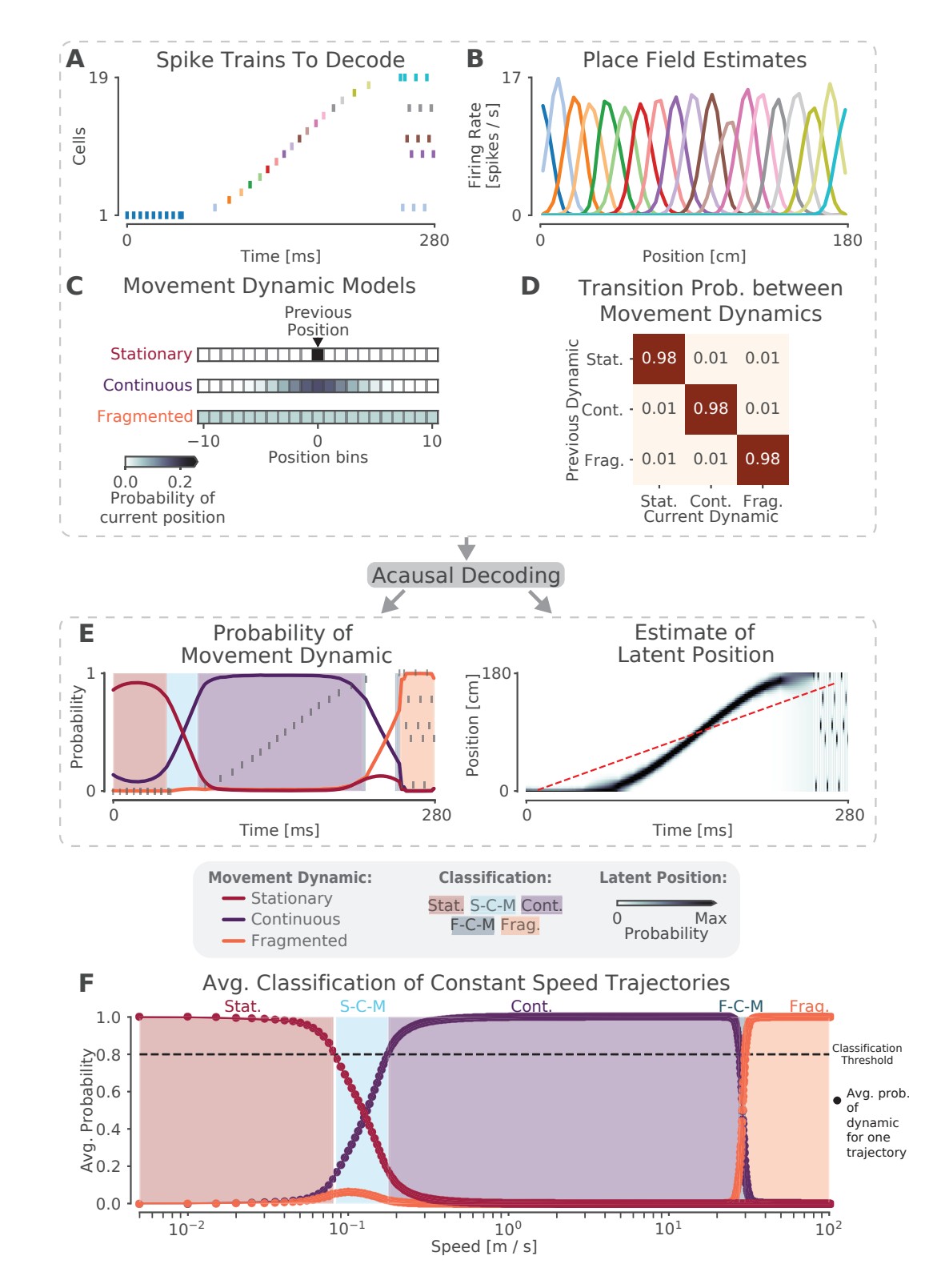

**Figure 1.** The model can capture different sequence dynamics on simulated data. (**A**) We construct a firing sequence of 19 simulated place cells that exhibits three different movement dynamics. For the first 60 ms, one cell fires repeatedly, representing one stationary location. For next 190 ms, the cells fire in sequence, representing a rapid continuous trajectory along the virtual track. For the last 30 ms, cells fire randomly, out of spatial order, representing an fragmented spatial sequence. (**B**) Like the standard decoder, the state space model uses estimates of cells' place fields from when the

*Figure 1 continued on next page*

*Figure 1 continued*

animal is moving and combines them with the observed spikes in (A) to compute the likelihood of position for each time step. (C) The prediction from the neural data is then combined with an explicit model of each movement dynamic, which determines how latent position can change based on the position in the previous time step. We show the probability of the next position bin for each movement dynamic model (color scale). Zero here represents the previous position. (D) The probability of remaining in a particular movement dynamic versus switching to another dynamic is modeled as having a high probability of remaining in a particular dynamic with a small probability of switching to one of the other dynamics at each time step. (E) The model uses the components in A-D over all time to decode the joint posterior probability of latent position and dynamic. This can be summarized by marginalizing over latent position (left panel) to get the probability of each dynamic over time. The shaded colors indicate the category of the speed of the trajectory at that time (Stat. = Stationary, S-C-M = Stationary-Continuous-Mixture, Cont. = Continuous, F-C-M = Fragmented-Continuous-Mixture, Frag. = Fragmented), which is determined from the probability. Marginalizing the posterior across dynamics also provides an estimate of latent position over time (right panel). Red dotted line in the right panel is the best fit line from the standard decoder using the Radon transform. (F) The probability of each dynamic depends heavily on the speed of the trajectory, as we show using a range of simulated spiking sequences each moving at a constant speed. Each dot corresponds to the average probability of that dynamic for a given constant speed sequence. We use a 0.80 threshold (dotted line) to classify each sequence based on the dynamic or dynamics which contribute maximally to the posterior (shaded colors).

The online version of this article includes the following figure supplement(s) for figure 1:

**Figure supplement 1.** The model is robust to change of the probability of persisting in the same dynamic for a wide range of plausible expected durations (25–150 ms).

compute an estimate of how each cell's firing rate varies over position during movement (i.e. the place field, *Figure 1B*). This is used during decoding to compute the Poisson likelihood of position over time given the spiking sequence of interest. In contrast to the standard decoding approaches, we can use small time bins (in our case 2 ms vs. 20 ms or more) because we are able to take advantage of the prior placed on the dynamics by the state space model. This allows us to detect changes on smaller time scales than would be possible with the standard decoder and incorporate information about times when there is no spiking (*Deng et al., 2015*). Further, because place estimates from spikes within a single bin are combined, our small time bins allow us to measure the spatial information conveyed by single spikes, rather than assuming that a downstream neuron would integrate spikes from multiple neurons on the timescale of 20 ms or more.

Next, we specify movement dynamic models that describe a variety of ways that the latent position—the 'mental' position of the animal represented by the cells— could evolve over time. (*Figure 1C*). We do this by defining a state transition matrix that defines how the latent position can move from the previous time step (in our case, 2 ms). Previous findings suggest that replay may exhibit at least three distinct types of movement dynamics: stationary (*Yu et al., 2017*; *Farooq and Dragoi, 2019*; *Muessig et al., 2019*), continuous (*Davidson et al., 2009*), and fragmented, which could correspond to both extended spatially incoherent representations and representations that jump from one position to another in a short time period (*Pfeiffer and Foster, 2015*). We therefore define movement dynamic models to capture each of these possibilities.

In the stationary movement dynamic, the latent position does not change between time steps. The state transition matrix can thus be defined as an identity matrix, which predicts that the next position will be the same as the last position (at the resolution of the position bin). In the continuous movement dynamic, the latent position is most likely to be 'spatially close' to the position in the previous time step, so we use a Gaussian random walk state transition matrix. This means that, for a given latent position, the probability of moving to another position is modeled by a Gaussian centered at that position and 'spatially close' is defined by the variance of the Gaussian. In our case, since replay has been shown to move at speeds much faster than the animal's speed (*Davidson et al., 2009*; *Pfeiffer and Foster, 2015*), we set the variance of the Gaussian to 6.0 cm. This ensures that with a 2 ms time step, the latent position is 95% likely to be within 4.90 cm of the previous latent position (or equivalently, this means that latent speeds from 0 to ~25 m/s are most likely). Last, in the fragmented movement dynamic, the latent position can move to any available position instantaneously. We model this using a uniform state transition matrix, which makes transitions to each position equally likely. Importantly, the fragmented dynamic provides a well-specified definition of what we mean by random, or unstructured activity.

Finally, we specify how likely each movement dynamic is to persist in time versus change to another dynamic via another state transition matrix between the movement dynamics (*Figure 1D*). In order to be conservative with respect to switching between dynamics, we assume that each

movement dynamic is likely to dominate for ~100 ms on average, which is approximately the duration of a SWR event. There is, however, a small probability of switching to one of the other movement dynamics. Accordingly, we set the probability of staying in a dynamic to 0.98 for each 2 ms time step, which corresponds to an expected duration of 100 ms for staying in a particular dynamic (the Markov assumption of the model means that the probability of staying in a dynamic follows a geometric distribution). Importantly, the data drives the estimated dynamic, so even if the probability of staying in a particular movement dynamic is high, data providing clear evidence for a change in dynamic can, within a time frame of ~10 ms, drive a change to a different estimated dynamic, as in our simulated example. In line with this, we show below that our results are relatively insensitive to the value of this parameter.

Once we have specified the data and the movement dynamic models, we have fully specified the state space model. We use acausal decoding, meaning that we use all information from both past and future spikes, to estimate the joint posterior probability of position and dynamic (see Materials and methods). With this, we can summarize the resulting posterior probability with two quantities: the probability of each movement dynamic over time (by integrating out position; *Figure 1E*, left panel) and the probability of latent position over time, irrespective of movement dynamic (by summing over the movement dynamics; *Figure 1E*, right panel).

An examination of the two summaries shown in *Figure 1E* reveals that that the model successfully captures the dynamics of the population spiking activity in *Figure 1A*. The stable firing of the one active neuron indicates a stationary representation, and accordingly, the probability of the stationary movement dynamic is high at the beginning of the simulated replay. A change in the data then drives a rapid transition to the continuous movement dynamic, reflecting the trajectory-like spatially sequential spiking from the population of cells. Subsequently, as the population activity becomes spatially incoherent, the fragmented movement dynamic dominates for the last 30 ms of the simulated event.

This illustrates two key features of the model. First, as mentioned above, the fragmented dynamic gives us a way to directly identify times when the position representation is spatially incoherent, a higher resolution and less assumption-dependent alternative to the more commonly used non-parametric shuffle. Second, this approach allows the model to to capture a wide range of movement speeds for the latent position in contrast to the standard decoder line fit (red dashed line in *Figure 1E*, using the Radon transform). The model is defined in terms of a mixture of movement dynamics, as summarized by the probability of each movement dynamic, and which dynamic or dynamics are dominant at a given moment is related to the temporal evolution of the underlying position representation. To demonstrate this, we applied the model to 10,000 simulated trajectories, each trajectory proceeding at a constant speed (*Figure 1F*) from 1 cm/s to 10,000 cm/s. From this, we can see that not only are there regions of speed that correspond to each of our three movement dynamics being highly probable (where we define highly probable to be greater than or equal to 0.80 probability), but there are also intermediate speeds where two of the dynamics exhibit relatively high probability; and where the sum of two of the dynamics' probabilities exceeds 0.80. In this manuscript, we will refer to these intermediate speeds as mixture dynamics. For example, when the stationary dynamic has a probability of 0.6 and the continuous has a probability of 0.4, we call this a stationary-continuous mixture (light blue, *Figure 1F*) and this indicates that the trajectory is moving slowly. Correspondingly, if the continuous dynamic has a probability of 0.5 and the fragmented dynamic has a probability of 0.4, then we would call this a fragmented-continuous-mixture and this indicates the trajectory is moving very quickly, but not as quickly as the fragmented dynamic dictates. In summary, we can characterize the speed or set of speeds that occur within an SWR based on the probability of each of the three movement dynamics over time. We further classify the probability of each movement dynamic as being part of one of five speed categories: stationary, stationary-continuous-mixtures, continuous, fragmented-continuous mixtures, and fragmented.

We note here that the choice of any particular threshold for classifying the movement dynamic of a SWR is arbitrary, and that the power of our approach lies in part on the ability to assign a probability for each dynamic or combinations of dynamics to each moment in time. Our goal in choosing 0.80 was to use a threshold that corresponds to highly probable events that roughly partition the latent position trajectory into interpretable categories of speed. Nonetheless, we also verify that our central results hold with a higher threshold of 0.95. We do not know if downstream neurons can explicitly responds to these dynamics based on either threshold, and there is probably no hard

boundary between these dynamics, but our approach makes it possible to ask that question in a systematic way.

Finally, as mentioned above, we wanted to test the robustness of the model to the choice of probability of staying in a dynamic, because our choice of 0.98 or an expected duration of 100 ms is only based on the expected duration of an SWR. To investigate this we decoded the spiking sequence in *Figure 1A* with different probabilities of staying in the same dynamic versus switching to another dynamic (*Figure 1—figure supplement 1*). We found that for a large range of plausible probabilities of staying in one of the dynamics (between 0.96 and 0.993, corresponding to an expected duration between 25 and 150 ms), the model identified the movement dynamics consistently, with high probability (*Figure 1—figure supplement 1A*), demonstrating that data itself is the most influential element in the model. Furthermore, the most probable latent positions remained relatively consistent across the range of these probabilities as well (*Figure 1—figure supplement 1B*).

## Identification of continuous dynamics in a SWR with sorted or clusterless spikes

We next sought to validate the model on real hippocampal data recorded from awake, behaving rats. To ensure that we could capture rapid trajectories many times the speed of the animal—as described by most studies of hippocampal replay—we first decoded a single SWR with sequential population activity (*Figure 2A*, top panel). Here the rat was performing a spatial alternation task on a W-shaped track (see Materials and methods for details). As expected, we observed that the probability of the continuous dynamic is high throughout this event, but the probability of the stationary dynamic was also noticeable at the beginning and end of the SWR (*Figure 2A*, middle panel). Using our speed classification scheme, this indicates that the speed of the replay is initially slower— a mixture of continuous and stationary dynamics—and then speeds up and slows down again. This is also evident in the posterior probability of latent linear position over time, which shows that the replay most likely travels down the center arm and up the right arm (*Figure 2A*, bottom panel). We can also see this when we project the maximum of the posterior of this trajectory (the most probable 'mental' position) to 2D to better see the spatial trajectory on the maze (*Figure 2C*). Importantly, when we apply the same model using the 2D position of the animal we get a similar result (*Figure 2—figure supplement 1A*).

One of our criteria for a more optimal method is for it to use all of the available spiking data. Using only clustered spikes discards any spike events that cannot be uniquely assigned to a putative single neuron, substantially reducing the amount of information that the resultant decoding can use. Additionally, spike sorting is not necessary to recover the underlying neural population dynamics (*Trautmann et al., 2019*) and often requires inclusion decisions that can vary widely across experimenters. We therefore adopted a 'clusterless' approach which directly uses multiunit spikes and their spike waveform features to decode position without spike sorting (see Materials and methods). Clusterless decoding has previously been used to successfully identify theta sequences and replay sequences in the hippocampus (*Chen et al., 2012b*; *Kloosterman et al., 2014*; *Deng et al., 2016*; *Kay et al., 2020*). Applying a clusterless decoder to the same SWR event, we get similar classification of the sequence (*Figure 2B,D*), both with 1D linearized position and 2D position (*Figure 2—figure supplement 1B*). As predicted, the spatial extent of the event is longer and the estimate of the posterior probability of latent position is narrower for the clusterless model. This reflects the clusterless model's access to a larger pool of data that provides more information about the extent of the event and more certainty in the latent position and the dynamic (*Figure 2D* vs C).

## Most replays contain coherent spatial content

After testing our model on a single SWR event, we applied our clusterless decoding algorithm to hippocampal recordings from 10 rats performing the W-track spatial alternation task (# tetrodes range: [10, 24], brain areas = [CA1, CA2, CA3]; some data previously used in *Karlsson and Frank, 2009*; *Carr et al., 2012*; *Kay et al., 2020*; position linearized to 1D). We first confirmed that our cell population for each animal could closely track the position of the animal during behavior by comparing the most probable decoded position to the position of the animal. We found that the average median difference between actual and decoded location is small (7 cm median difference, 5–9 cm 95% CI, for all times where the animal was moving greater than 4 cm/s, five-fold cross validation).

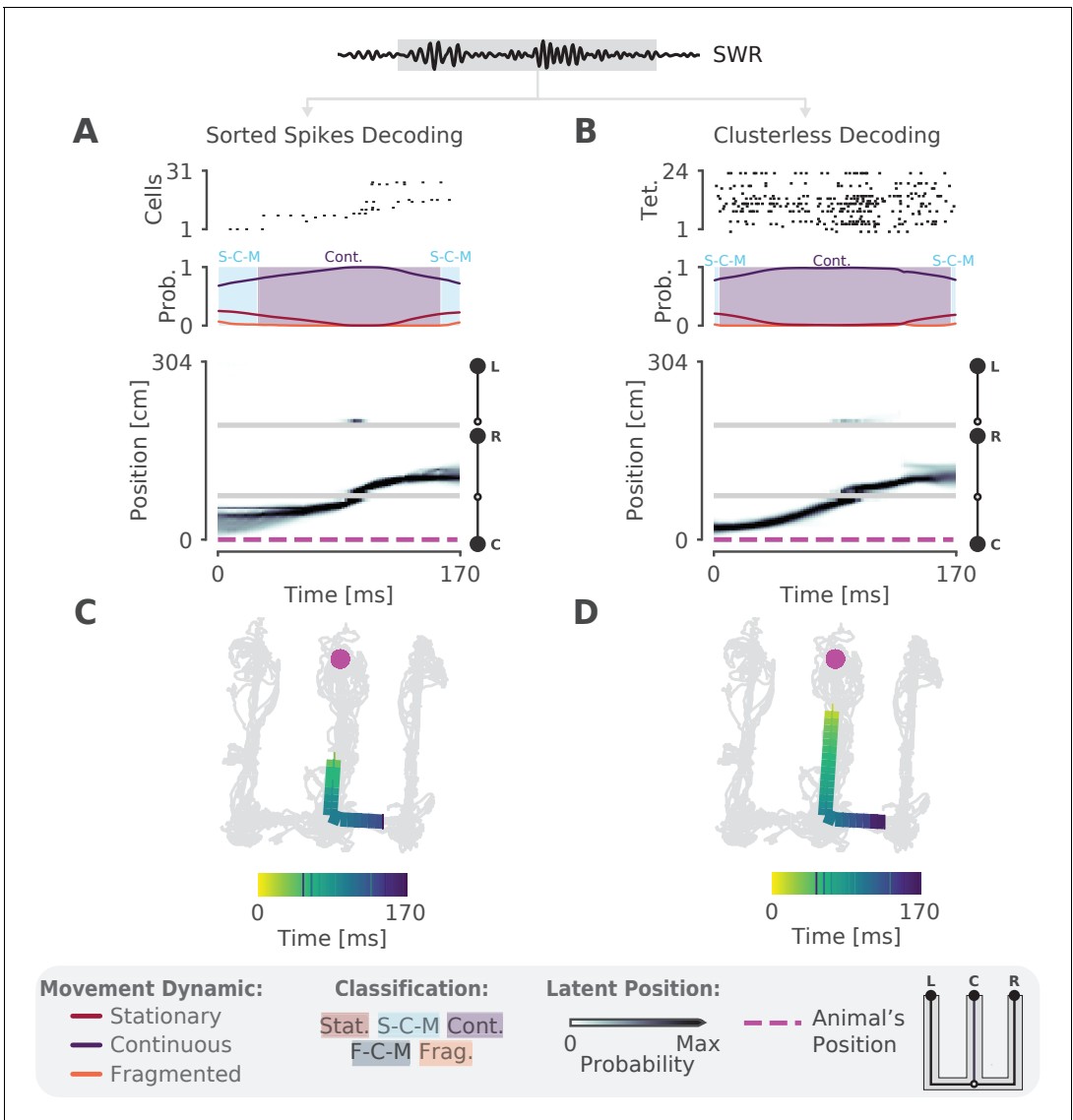

**Figure 2.** The model can decode hippocampal replay trajectories using either sorted and clusterless spikes from the same SWR event. (**A**) Decoding using sorted spikes. The top panel shows 31 cells on a W-track ordered according to linearized position by their place field peaks. The middle panel shows the probability of each dynamic over time as in *Figure 1E*, left panel. Shaded regions correspond to the speed classifications as in *Figure 1F*. The bottom panel shows the estimated probability of latent position over the course of the SWR as it travels down the center arm toward the right arm. L, R, C correspond to the position of the left, right and center reward wells, respectively. The animal's actual position is indicated by the the magenta dashed line. Underneath is the maximum of the 1D decoded position (the most probable position) projected back onto the 2D track for the sorted decoding. Color indicates time. The animal's actual position is denoted by the pink dot. Light gray lines show the animal's 2D position over the entire recording session. (**B**) Decoding using clusterless spikes. The top panel shows multiunit spiking activity from each tetrode. Other panels have the same convention as (**A**). Underneath is the maximum of the 1D decoded position (the most probable position) projected back into 2D using the clusterless decoding.

The online version of this article includes the following video and figure supplement(s) for figure 2:

**Figure supplement 1.** Decoding the same SWR in *Figure 2* with 2D position using sorted spikes and clusterless decoding.

**Figure supplement 2.** More examples of SWRs that have continuous trajectories.

**Figure supplement 3.** Population firing rate on the track is spatially uniform and consistent for each animal.

**Figure 2—video 1.** Example of an SWR with continuous content.

https://elifesciences.org/articles/64505#fig2video1

This level of accuracy is comparable to that of other studies (*Davidson et al., 2009*; *Shin et al., 2019*; *Farooq and Dragoi, 2019*), and reflects in part the presence of 'theta sequences' (*Dragoi and Buzsáki, 2006*; *Foster and Wilson, 2007*) where position deviates behind and ahead of the animal across individual theta cycles. Importantly, we achieved this using small time bins (2 ms) compared to the commonly used 250 ms time bin that most studies use for decoding during movement, demonstrating the power of our algorithm. For each animal, we also verified that the population spiking rate was relatively consistent over all positions on the track, suggesting that we have comparable sampling of all locations in the environment (*Figure 2—figure supplement 3*).

Having established the validity of our decoder in tracking the animal's position, we next assessed the prevalence of spatial content across SWRs. We detected SWRs using a permissive threshold (see Materials and methods) to ensure that we include both the rare large-amplitude events as well as the much more common small-amplitude events. As expected, and providing further support for the validity of the model, we observed replays that are classified as continuous throughout the entirety of the SWR, similar to those seen using standard approaches (*Figure 2—figure supplement 2A–F*). However, we also observed many events with spatially coherent content that do not have this structure. For example, there are trajectories that start in one direction and reverse back to the original position (*Figure 3A*, *Figure 3—figure supplement 1B*), representations that remain fixed in one position (*Figure 3B*, *Figure 3—figure supplement 1G*), and trajectories that jump between arms and between dynamics (*Figure 3C,D,F*, *Figure 3—figure supplement 1F,H,I*). We also observed SWRs where the content is spatially incoherent throughout (*Figure 3—figure supplement 1A,D*).

Using a 0.80 threshold, 89% (23,071 of 25,844) of SWRs contain at least one of the three dynamics or dynamic mixtures. To ensure that this reflects the spatially tuned firing of the neurons, we trained the encoding model with positions resampled with replacement, a shuffling procedure which disrupts the relationship between spiking and position, for two recording sessions. We then decoded the same SWR events, containing the original spikes. Only 9% of the SWRs are classified in the shuffled datasets, a value that is significantly less than that seen for the real data (p=0.02 for recording session 1, p=0.02 for recording session 2, *Figure 3—figure supplement 2*). This shows that our model does not impose movement dynamics in the absence of spatial information in the data.

Previous work focusing on spatially sequential replay reported that only a minority of events contain sequential spatial content (*Foster and Wilson, 2006*; *Karlsson and Frank, 2009*; *Davidson et al., 2009*). We therefore asked what fraction of classified events contain spatially coherent content, which we define as any times with stationary, stationary-continuous mixture, or continuous dynamics (see Materials and methods and *Figure 1F*). We find that 96% (22,170 of 23,071) of classified SWRs and 86% (22,170 of 25,844) of all SWRs include spatially coherent structure, and that this prevalence of spatially coherent structure is consistent across animals (*Figure 3G*). We then asked what fraction of events contained spatially incoherent content, defined as an SWR containing any times with fragmented or fragmented-continuous mixture dynamics. We find that only 14% (3295 of 23,071) of classified SWRs include any times with spatially incoherent structure (*Figure 3G*). To further validate this result, we performed a shuffling procedure that, in contrast to our previous shuffle, preserves the local correlations between spikes but reduces global spatial structure. This controls for dynamics that may have been induced by bursts or other local features of the spike train while randomizing the position to spike relationship. To do this, we randomized the order of runs (from one reward well to another) and then circularly permuted the resulting segments of data across all tetrodes uniformly. Using this shuffle, we found that there was still less spatially coherent and more spatially incoherent content compared to the real data (recording session #1, spatially coherent: 99% vs. 71%, p=0.02, real vs. shuffled; spatially incoherent: 22% vs. 61%, p=0.02, real vs. shuffled; recording session #2, spatially coherent: 95% vs. 63%, p=0.02, real vs. shuffled; spatially incoherent: 16% vs. 79%, p=0.02, real vs. shuffled; *Figure 3—figure supplement 3*).

To more directly compare our findings to previous work, we quantified the percentage of classified SWRs that contained continuous content, as would typically be analyzed when using the standard decoder. Here, our findings were consistent with previous reports: in our datasets, 4449 of 23,071 or 19% of classified SWRs included time periods where the decoded position was classified as continuous (*Figure 3G*, 17% of all SWRs). Thus, focusing on only high-speed, linear-fit trajectories excludes a large fraction of events where there is evidence for spatially coherent, but not continuously changing, content. We emphasize here that we did not limit our analysis to only those SWRs

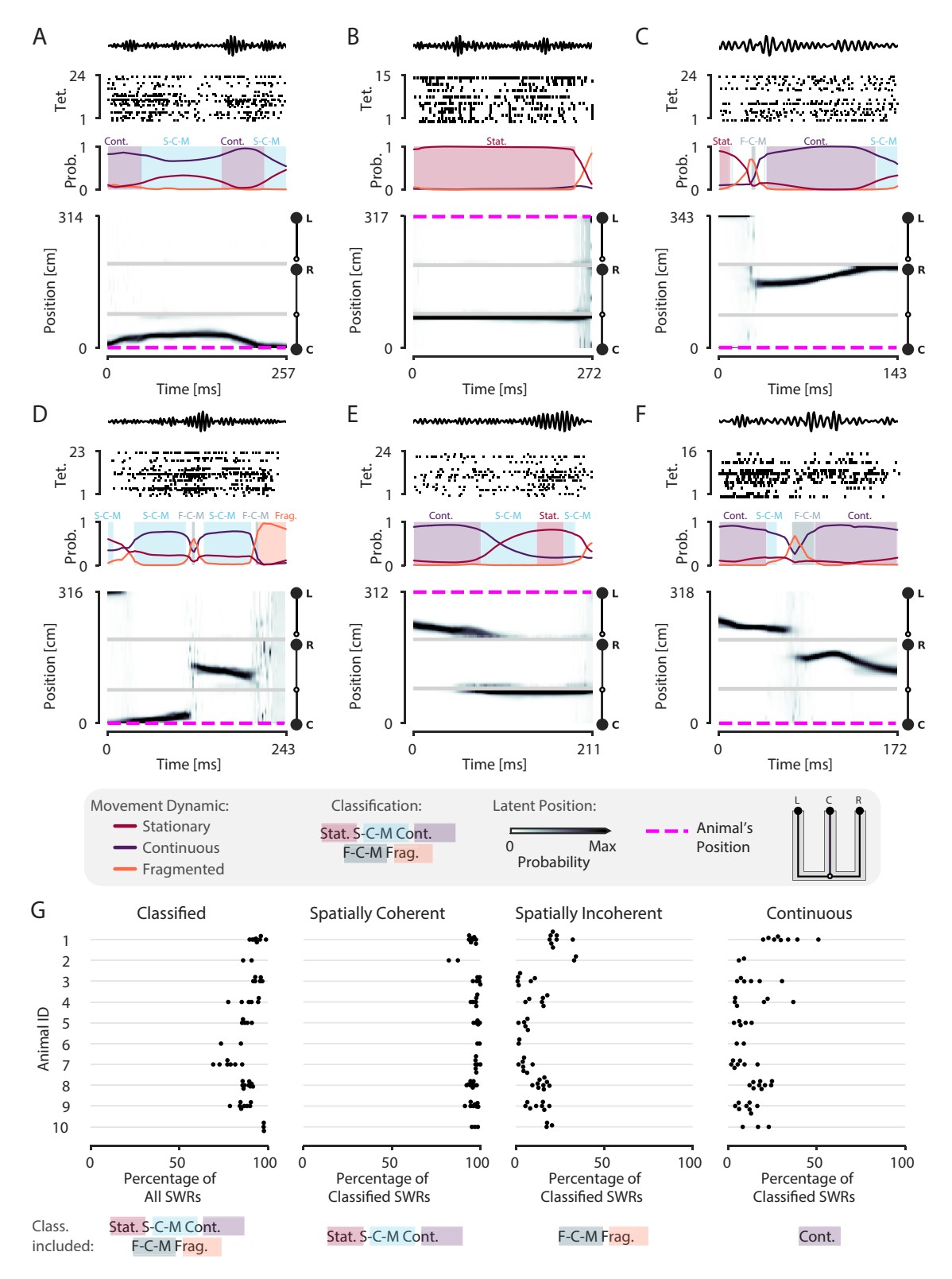

**Figure 3.** Most SWRs are spatially coherent, but not continuous. (A-F) Examples of SWRs with non-constant speed trajectories. Figure conventions are the same as in *Figure 2*. Filtered SWR ripple (150–250 Hz) trace from the tetrode with the maximum amplitude displayed above each example. (A) An SWR where the decoded position starts moving down the center arm away from the animal's position at the center well, slows down, and returns back. (B) An SWR where the decoded position persistently stays at the choice point (open circle) while the animal remains at the left well. (C) An SWR where

*Figure 3 continued on next page*

*Figure 3 continued*

the decoded position begins with stationary representation of the left well, then jumps to the middle of the right arm and proceeds up the right arm to the right well. (**D**) An SWR where the decoded position begins with stationary representation of the left well, jumps to the center arm, proceeds away from the center well, jumps to the right arm, proceeds back toward the center well, and then becomes fragmented. (**E**) An SWR where the decoded position begins in the left arm and persists at the end of the center arm. (**F**) An SWR where the decoded position starts in the left arm toward the choice point, jumps to the right arm and proceeds back toward the choice point. (**G**) Classification of SWRs from multiple animals and datasets. Each dot represents the percentage of SWRs for each day for that animal. An SWR is included in the numerator of the percentage if at any time it includes the classifications listed below the column. The denominator is listed in the x-axis label.

The online version of this article includes the following video, source data, and figure supplement(s) for figure 3:

**Source data 1.** Table of replay statistics for each SWR for *Figure 3G*.
**Figure supplement 1.** More examples of SWRs with non-constant speed trajectories.
**Figure supplement 2.** Shuffling the position data with replacement decreases the percent of SWRs classified.
**Figure supplement 3.** Shuffling the data by swapping the runs and circularly permuting the position increases the percentage of spatially incoherent SWRs and decreases the spatially coherent SWRs.
**Figure 3—video 1.** Example of an SWR with that is not purely continuous.
https://elifesciences.org/articles/64505#fig3video1

that had continuous dynamics for the entire SWR, as is assumed by line-fitting approach of the standard decoder. The consideration of this broader class of SWRs allows us to take advantage of the ability of our decoder to capture a range different speeds within each SWR event.

We repeated our classification analysis with a higher classification threshold of 0.95 to ensure that our result was not dependent on the threshold of 0.80. We find that, while this change slightly reduced the total fraction of classified SWRs (19,478 of 26,159 or 74% of all SWRs), an even higher fraction of the classified SWRs (19,317 of 19,478 or 99% classified SWRs) included spatially coherent content. Similarly, SWRs containing spatially incoherent content comprised a small fraction of the classified SWRs (490 of 19,478 or 3% classified SWRs).

Because our model is specified in the context of a latent position associated with different movement dynamics, it allows us to not only classify events in terms of their dynamics, but also to quantify the model's certainty in each position estimate at each moment in time given the model parameters. To do so, we can compute the cumulative spatial size of the 95% highest posterior density (HPD) region of the latent position estimate. The 95% HPD region corresponds to the set of most probable latent positions as estimated by the model. Larger values of the HPD region size indicate the model is less certain about position, because the most probable decoded positions are distributed over more of the track at a given time point. In contrast, smaller values of the HPD region size indicate that the model is more certain about the estimate of position because the extent of the HPD is more concentrated and covers less of the track. Thus, the HPD region size provides a complementary measure of spatial information, and evaluating it allows us to verify that the events we defined as spatially coherent also correspond to events where there is high certainty around the position estimates.

We find that spatially coherent events indeed have smaller HPD regions than events with fragmented dynamics. *Figure 4A* and *Figure 4B* show two example SWRs that are classified as having continuous and stationary dynamics, respectively. The most probable positions (as estimated by the model) at each time step in these SWRs is concentrated in a small portion of the track and correspondingly, the HPD region is small throughout the SWRs. In contrast, *Figure 4C* shows a SWR where the dynamics are fragmented and correspondingly, the most probable positions are more spatially diffuse and the HPD region is much larger. The HPD region size also provides insights into the moment-by-moment structure of each event, which can change over the time course of a SWR. An example of this change is shown in *Figure 4D*, where the HPD region is small for most of the SWR until the end, at which point the uncertainty of position becomes much higher, reflecting a transition from a spatially coherent to a spatially incoherent representation. Overall, when we examine the average HPD region size for each SWR, grouped by dynamic, we find a clear bimodal split between spatially coherent dynamics and spatially incoherent dynamics (*Figure 4E*). For the spatially coherent dynamics, the average HPD region for each SWR was much smaller than the spatially incoherent dynamics (median 24 cm, spatially coherent vs. median 238 cm, spatially incoherent, p=2.2e-16, one-sided Mann-Whitney-U). The HPD region for the unclassified SWRs was similarly large.

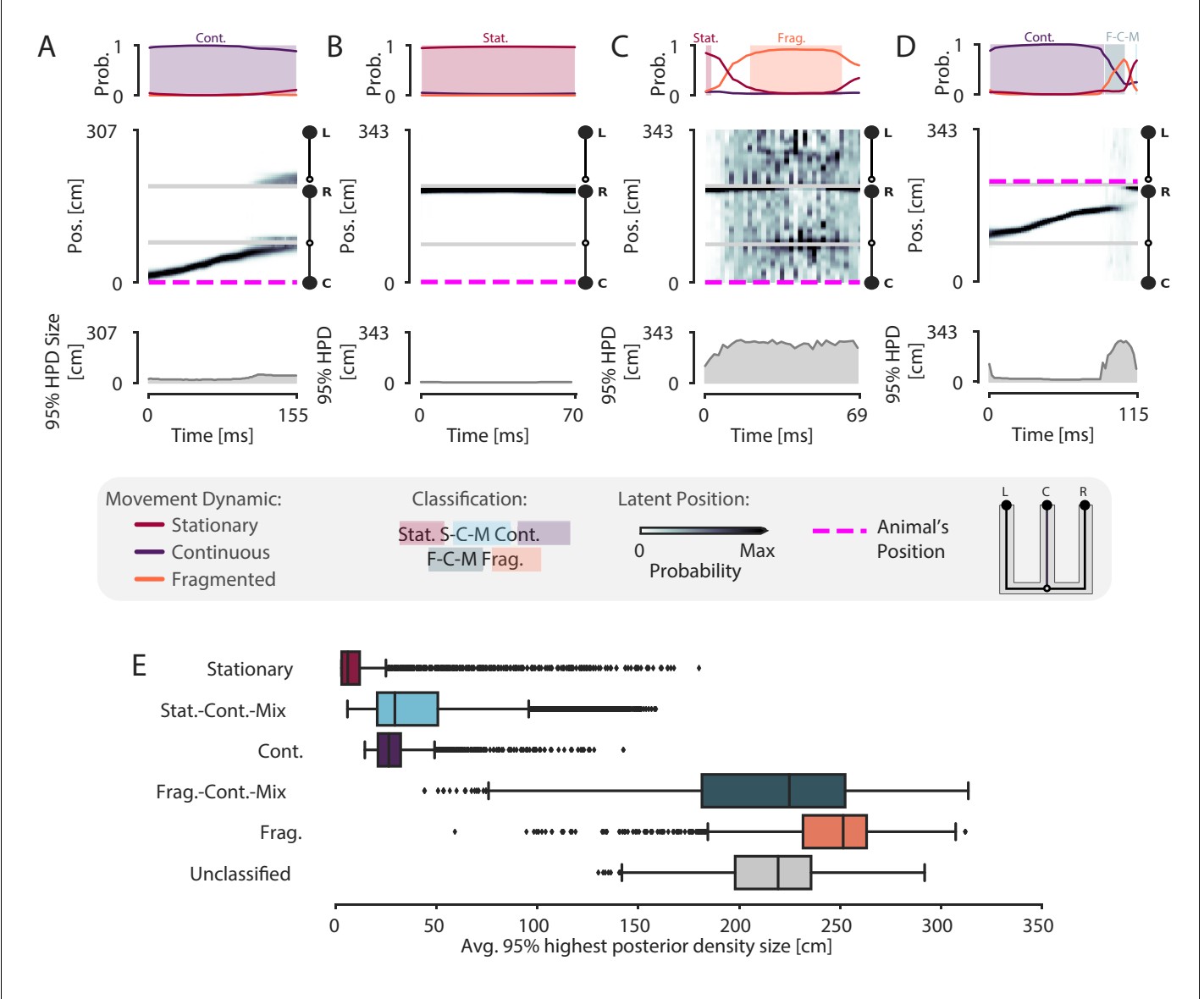

**Figure 4.** Validation of classification using the 95% Highest Posterior Density. (**A–D**) Examples of the 95% Highest Posterior Density. In each column: top panel: Probability of dynamic over time. Shading and labels indicate dynamic categories. Middle panel: Posterior distribution of estimated linear position over time. Magenta dashed line indicates animal's position. Bottom panel: HPD region size—the integral of the position bins with the top 95% probable values. (**E**) Average 95% HPD region size for each dynamic category. Median 24 cm for spatially coherent vs. median 238 cm for spatially incoherent, p=2.2e-16, one-sided Mann-Whitney-U.

The online version of this article includes the following source data and figure supplement(s) for figure 4:

**Source data 1.** Table of replay statistics for each SWR for *Figure 4E*.

**Figure supplement 1.** Shuffling the position with replacement increases the average 95% HPD region size.

**Figure supplement 2.** Shuffling the position data by swapping the runs and circularly permuting the data increases the average 95% HPD region size for spatially coherent and incoherent classified times.

We note here that while the size of the HPD region will be influenced by the dynamic estimated from the data, it remains possible for continuous or stationary dynamics to correspond to a large HPD region (see outliers in *Figure 4E*), indicating less spatial certainty for those specific events. In general, if the data does not exhibit strong place specific firing, the HPD region will be large, regardless of dynamic classification. To show this, we used the same shuffle as above, resampling position with replacement for two recording sessions and shuffling the relationship between spiking

and spatial information when fitting the encoding model. The shuffled decodes have much larger HPD regions than the real data (recording session #1: 55 cm vs. 229 cm, p=0.02, real vs. shuffled; recording session #2: 72 cm vs. 231 cm, p=0.02, real vs. shuffled, one-sided; *Figure 4—figure supplement 1*). We also compared the HPD regions between the real data and a shuffle that swapped the runs and circularly shuffled position, as above. This also resulted in larger HPD regions than the real data for spatially coherent and incoherent SWRs (recording session #1, spatially coherent: 29 cm vs. 52 cm, p=0.02, real vs. shuffled; spatially incoherent: 181 cm vs. 243 cm, p=0.02, real vs. shuffled; recording session #2, spatially coherent: 26 cm vs. 53 cm, p=0.02, real vs. shuffled; spatially incoherent: 210 cm vs. 263 cm, p=0.02, real vs. shuffled; *Figure 4—figure supplement 2*).

## Many replay events are slow or stationary

Surprisingly, our results indicate that most events with coherent spatial content are not dominated by the continuous movement dynamic, but instead correspond to trajectories that are stationary or that move relatively slowly compared to their purely continuous counterparts. We therefore examined these events in more detail. We first note that most of the SWRs (14,989 of 21,433 or 70% of classified SWRs) were classified as having only a single dynamic or dynamic mixture (*Figure 5A*, for specific example SWR see *Figure 3B*), whereas SWRs with multiple dynamics or dynamic mixtures (such as those in *Figure 3A,C,D,E and F*) were less common. Interestingly, the most common category of classified SWRs were those with only stationary-continuous mixtures (8944 of 21,433 or 42% of classified SWRs, *Figure 5A*, *Figure 5—figure supplement 1*, note that this percentage ignores the unclassified category). These events contain representations that move at slower speeds (*Figure 5D*)—similar to those speeds experienced by the animal in the environment (*Figure 5G*, median 17 cm/s for run periods and 4 cm/s for all times)—and are sustained, but slightly shorter, on average, than events with continuous dynamics (median duration: stationary-continuous-mixture 73 ms vs. continuous 94 ms, *Figure 5B*). Nonetheless, both the slow-moving events and continuous events frequently represent locations that were some distance away from the animal (mean trajectory distance from animal's position: stationary-continuous-mixture 52 cm vs. continuous 43 cm, *Figure 5C*). This indicates that the content of these SWRs, like those that are classified as continuous, do not represent the animal's actual position. We note that roughly a third of these events persisted for the entire duration of the SWR, but the other two thirds included some time where the model was uncertain about the dynamic. This shows the power of our state space decoder because it allows us identify periods of time when the model has high confidence with each SWR, and use these periods to characterize the SWR overall or to focus on specifically for further analysis.

The second most prevalent classification was exclusively stationary events (4858 of 21,433 or 23% classified SWRs; 47% of SWRs with stationary dynamics were entirely stationary for the entire duration of the SWR). Unlike the stationary-continuous mixtures, most of these events represented a location close to the animal's actual position (*Figure 5C*). There were, however, a number of stationary events that represented positions far from the animal. We set a threshold of 30 cm distance from the animal to identify non-local stationary events and found such content in ~7% of classified SWRs (1437 of 21,433 classified SWRs). Of these, 46% (664 of 1437 stationary SWRs) were stationary throughout the duration of the SWR. These non-local stationary events most commonly represented reward wells or choice points (*Figure 5E*), consistent with *Yu et al., 2017*, but a small portion of them occurred at other locations of the track. This suggests that these representations can be flexibly deployed to represent discrete locations.

## Control analyses

We then carried out a series of control analyses to determine whether our results could be due to differences in spiking statistics across dynamics or interneuron-specific activity patterns. We first calculated the multiunit spiking rates and found that these were similar across the different dynamics (*Figure 5F*). This indicates that all dynamics, including stationary and stationary-continuous mixtures, were driven by comparable levels of sustained spiking information and could not be explained by the absence of spiking. Further corroborating this, we found that when we re-ran our analysis using periods of high multiunit activity instead of SWRs to identify events, we still found a larger proportion of stationary and stationary-continuous dynamics relative to the continuous dynamics (*Figure 5— figure supplement 3F*).

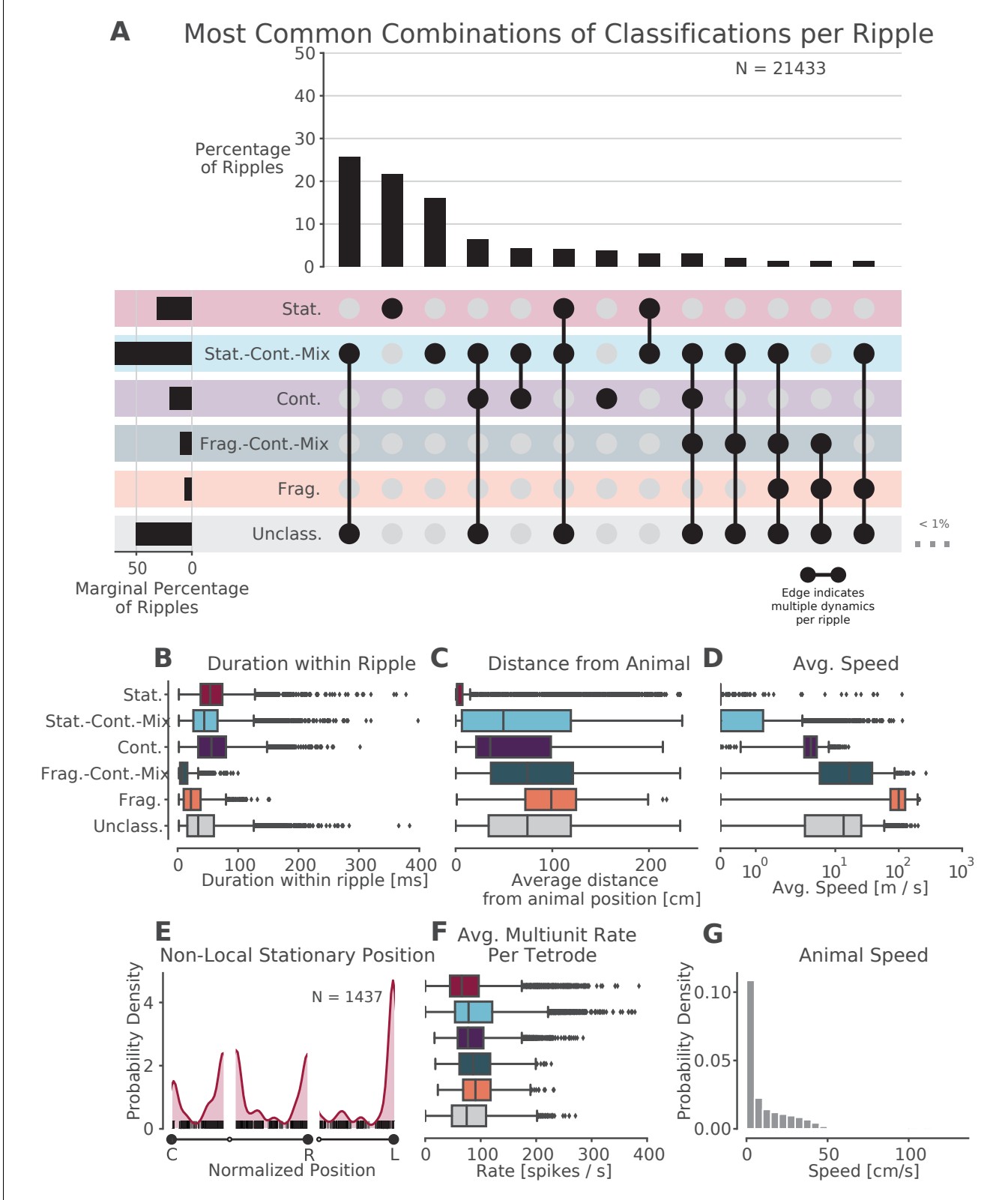

**Figure 5.** Prevalence of classifications. (**A**) UpSet plot (*Lex et al., 2014*)—which is similar to a Venn diagram with more than three sets—of the most common sets of classifications within each SWR. Each row represents a classification and each column represents a set of classifications, where filled-in black dots with an edge between the dots indicates that multiple classifications are present in the SWR (at different times). The sets are ordered by how often they occur as indicated by the bar plot above each category. The total percentage of each classification is indicated by the rotated bar plot on

*Figure 5 continued on next page*

*Figure 5 continued*

the left. (B) Duration of each dynamic category within a SWR. The box shows the interquartile range (25–75%) of the data and the whiskers are 1.5 times the interquartile range. Outlier points outside of this are labeled as diamonds. (C) Average distance between the latent position and the animal's actual position for each classification within the SWR. (D) Average speed of the classification within the SWR (excluding classifications with durations less than 20 ms). Note that these speeds are calculated using the most probable position (MAP estimate) which can be noisy when the probability of position is flat or multimodal.(E) Kernel density estimate of the position of stationary trajectories on the W-track at least 30 cm away from the animal's position. The shaded region represents the density estimate while the black ticks represent the observed non-local stationary positions. (F) Average tetrode multiunit spike rates for each dynamic category within each SWR (excluding classifications 20 ms). (G) Probability density of animal movement speeds, illustrating prevalence of slower speed real-world movement consistent with stationary and stationary-continuous mixture replay events.

The online version of this article includes the following video, source data, and figure supplement(s) for figure 5:

**Source data 1.** Table of replay statistics for each SWR for *Figure 5*.
**Figure supplement 1.** More examples of stationary-continuous-mixtures.
**Figure supplement 2.** Control analyses for distribution of dynamics.
**Figure supplement 3.** Further quantification of spiking and ripple properties by dynamic.
**Figure 5—video 1.** Example of a stationary-continuous-mixture.
https://elifesciences.org/articles/64505#fig5video1

---

We also verified that our classifications were very unlikely to be driven by the firing of interneurons, which are less likely to project to other brain areas and exhibit less spatial specificity than principal neurons in the hippocampus (*Wilent and Nitz, 2007*; *Jinno et al., 2007*; *Hangya et al., 2010*). First, we increased our tetrode participation threshold from two up to five tetrodes active during the SWR, to eliminate the possibility that events with a single tetrode with a high rate interneuron might drive the classification of stationary or stationary-continuous mixture dynamics during SWRs. We found that our results were robust to this increase in threshold (*Figure 5—figure supplement 2A*). Indeed, most of our events and most periods of dynamics within each event included spikes from upwards of 10 tetrodes (*Figure 5—figure supplement 3*). Next, we removed spikes that had spike widths of less than 0.3 ms to reduce the potential contribution of narrow-waveform inhibitory interneurons (*Fox and Ranck, 1975*). Similarly, this had little effect: we still observed a high proportion of SWRs with stationary and stationary-continuous mixture dynamics (*Figure 5—figure supplement 2B*). Using clustered data, we can more robustly categorize units as putative interneurons; thus, we then repeated this analysis on clustered data for 9 of the 10 animals, excluding putative interneurons based on spike width and firing rate and requiring that at least three putative pyramidal cells be active during the SWR. Although there were many fewer events to examine, we still observed many stationary and stationary-continuous mixtures in SWRs (*Figure 5—figure supplement 2D*). Finally, if interneurons with less spatial specificity were driving the classification of slower dynamics, we would expect to see the posterior be less spatially concentrated during these events. Instead, we found that stationary and stationary-continuous mixtures tended to have narrow posteriors (as characterized by 95% HPD region size in *Figure 4E*). This provides further evidence that our large fraction of slow-speed events were not the solely the result of the firing of interneurons.

We then asked whether the slow spatially coherent events (stationary-continuous mixtures) potentially reflect a slow theta sweep rather than a replay event. We noted that this seems very unlikely given that the stationary-continuous mixtures tended to represent locations far from the animal that would require rapid, long-distance theta sequences (median 50 cm *Figure 5C*). In addition, the majority of our events occur when the animal is moving at speeds less than 2 cm/s, as expected for SWRs (*Figure 5—figure supplement 3B*). Further, when we restrict the analyses to SWRs where the animal was moving at speeds less than 1 cm/s, we find similar proportion of SWRs with stationary and stationary-continuous mixture dynamics (*Figure 5—figure supplement 2C*).

A third possible concern is that burst firing from a single cell is driving the stationary dynamics. There are several lines of evidence against this. First, as mentioned above, most of our SWRs contained spikes from multiple tetrodes, including those with stationary dynamics (*Figure 5—figure supplement 3A*), so it is unlikely that a single cell could drive these dynamics. Second, the stationary dynamics have a longer duration than one would expect from a burst alone (*Figure 5B*, 54 ms median duration, 38–74 ms quartiles vs. bursts which have a typical duration of 6–24 ms; *Ranck Jr, 1973*; *Harris et al., 2001*; *Tropp Sneider et al., 2006*). Third, only ~50% of the spikes had interspike intervals (ISIs) of less than 6 ms during the stationary, stationary-continuous, and continuous

dynamics (*Figure 5—figure supplement 3A*), indicating that rapid bursts made up at most half of the observed spikes, and likely fewer given that these ISIs are computed for each tetrode, rather than for single units.

Finally, we wished to assess whether the spiking information from different hippocampal subfields (CA1, CA2, and CA3), could influence the types of dynamics we observed. This seemed unlikely, given that CA1 and CA3 spiking is tightly coordinated within and across hemispheres during SWR replay (*Carr et al., 2012*) and multiple previous papers have combined recordings across subfields and obtained results similar to those that restricted their analyses to CA1 (*Diba and Buzsáki, 2007*; *Karlsson and Frank, 2009*). However, we verified that when we used only CA1 tetrodes to decode (*Figure 5—figure supplement 2E*), we get similar proportions of dynamics as when using all the tetrodes.

## Comparison to standard approaches

Finally, we asked how our method compares to the standard approaches to identifying significant replay events that use a non-parametric shuffle test to characterize their content. This comparison is complicated due to the variety of parameters and inference methods used previously, so here we chose three commonly used approaches and a reasonable parameter set for the comparison. We began with an examination of events based on statistical significance of a line fit to a circularly per-muted spatial distribution, which has been the standard in the field. We applied the clusterless algo-rithm in 20 ms bins to produce probability densities of represented locations, as this binning is comparable to previous methods. We then used the Radon transform method, which finds the line with the highest summed posterior probability of position along the line (*Davidson et al., 2009*; *Farooq and Dragoi, 2019*), to calculate, for each SWR, a measure of significance based on the assumption that the probability density moves at a constant speed throughout the event (see exam-ples in *Figure 6—figure supplement 1A–F*, fourth row, blue line). We found that ~12% of the classi-fied events (as defined by our state-space decoder) were identified as significant by this Radon method. Not surprisingly, the fraction of significant events was strongly related to which dynamic they contained. Of the SWRs that contained a period of continuous dynamics, 31% (1386 of 4511 SWRs) were significant. Of all SWRs that contained a period of stationary dynamics, 8% (542 of 7176 SWRs) were significant, and when we restricted the analysis to long stationary events (>100 ms dura-tion), we found that 28% were significant. This increase is not surprising, since the shuffle used for the Radon method penalizes shorter events. An alternative approach, the linear regression method, which samples from the posterior probability and uses linear regression to fit a line to the samples (*Karlsson and Frank, 2009*; *Carr et al., 2011*; *Shin et al., 2019*), detects only non-zero slopes as significant, and thus by definition cannot identify stationary events as significant.

Conclusions about the representational content of events were similarly variable across methods. In addition to the Radon and linear regression methods, we also computed the MAP estimate, the maximum probability position of each 20 ms time bin (see examples in *Figure 6—figure supple-ment 1A–F*, fourth row, green line, *Pfeiffer and Foster, 2015*; *Stella et al., 2019*; *Kaefer et al., 2020*). We then compared the speed of the replay for each dynamic and for the entire SWR (*Fig-ure 6*). We found that the methods that rely on fitting a line for the entire SWR (the Radon transform and the linear regression), tend to estimate that the replays proceed at faster speeds (*Figure 6A,C*), whereas the MAP method (*Figure 6B*), which only relies on picking the most probable position for a given time bin and allows for more variable speeds, meaning it can capture both fast and slow speeds. In addition, the speeds from the MAP method (*Figure 6B*) are quite similar to the speeds estimated by the state space decoder (*Figure 6D*). This is to be expected because the MAP method imposes less smoothing (although some is added by using 20 ms time bins, see *Figure 6—figure supplement 2A–F* for the same examples with 2 ms bins), whereas the linear methods impose a larger amount of smoothing and cannot change directions (see *Figure 6—figure supplement 1A* for an example where this is beneficial because the trajectory is linear and the smoothing ignores a lack of spiking and *Figure 6—figure supplement 1B* for an example where this is not beneficial because the linear fit cannot change direction). Our state space decoder with 2 ms bins imposes much less smoothing than the linear methods and therefore is closer to the MAP estimates with 20 ms bins (*Figure 6—figure supplement 1B*), but also takes into account the uncertainty of each time bin (*Fig-ure 6—figure supplement 1C*) and is able to change direction more quickly than the 20 ms time bins (*Figure 6—figure supplement 1E*). Using 2 ms bins with the MAP method would allow for

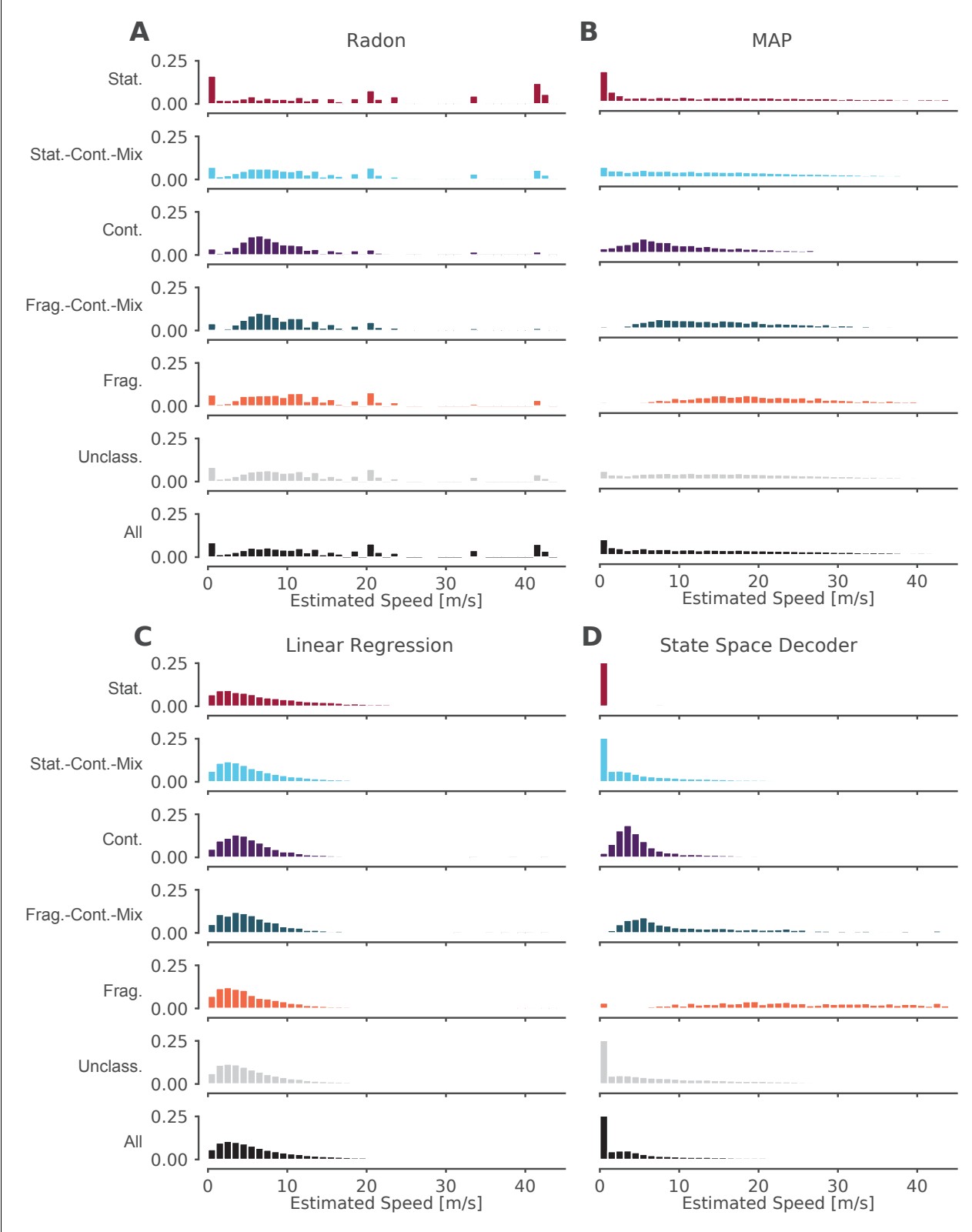

**Figure 6.** The standard decoder MAP estimate speeds are the most similar to the state space decoder. Event speeds calculated using three common 'Bayesian' decoder approaches (**A–C**) compared to using the state space model (**D**). For each panel, the top five rows show the probability density of the estimated speed for SWRs that contain that dynamic or combination of dynamics. Note that these categories are not mutually exclusive since a

*Figure 6 continued on next page*

*Figure 6 continued*

SWR can include more than one dynamic or combination of dynamics. The sixth row shows all SWRs containing unclassified dynamics. The final row is the estimated speed for all SWRs. (**A**) Radon transform (**B**) MAP estimate (**C**) Linear regression (**D**) State space decoder presented in this manuscript.

The online version of this article includes the following source data and figure supplement(s) for figure 6:

**Source data 1.** Table of replay statistics for each SWR for *Figure 6*.
**Figure supplement 1.** Examples of fits from the standard 'Bayesian' decoder with 20 ms bins and state space model.
**Figure supplement 2.** Examples of fits from the standard 'Bayesian' decoder with 2 ms bins and state space model.

faster change in the direction of the latent position but also results in much more variable estimates (*Figure 6—figure supplement 2*). These results indicate that the methods that made more assumptions about the structure of replay (Radon and linear regression approaches) yield different conclusions about which events are potentially meaningful and how those events progress over time from methods that make fewer assumptions (the MAP and state-space approaches). The substantial variability in both the proportions of 'significant' events and the speed of the events demonstrates that key conclusions about the events could be very different depending on the which of the previous methods and parameter were used.

## Discussion

We developed a state space model that identifies and quantifies replay in terms of a mixture of three movement dynamics: stationary, continuous, and fragmented. This model is robust: it is relatively insensitive to a range of plausible transition parameters between dynamics and instead strongly reflects the hippocampal place representation structure. We show that this model is interpretable: it has well-defined parameters that capture intuitive movement models and allows us to explain a large fraction of SWRs in terms of these dynamics—far more than have been previously associated with identifiable activity patterns. This model is also flexible: it can be used with sorted or clusterless spikes, it can be used with 1D or 2D positions, and—because it uses a mixture of dynamics—it allows us to confidently discover not only SWRs with spatial representations that move at a constant speed, but also representations that vary in speed, have slower speeds, or that are stationary. In fact, these slower moving representations constitute the majority of representations seen during SWRs, but would not be found with the typical approaches that assume that events progress at constant speed. The prevalence of these events indicates that hippocampal replay can recapitulate experience at both accelerated and closer to real-world speeds.

Previous work reported that less than half of SWRs or high-population-activity events contained replay events (*Foster and Wilson, 2006*; *Karlsson and Frank, 2009*; *Davidson et al., 2009*). This could lead one to the assumption that most events are not spatially coherent or meaningful. Our results indicate that this is not the case: in our dataset, the large majority of SWRs (86%) contained spatially coherent content, as defined by some combination of stationary and continuous dynamics. These events were mostly decoded representations where the most probable positions were highly spatially concentrated (i.e. had small HPD region size), indicating that they were driven by spatially specific firing. This was in contrast to unclassified events or events that included fragmented dynamics, which were associated with highly distributed decoded spatial representations.

Importantly, the spatially coherent events most often expressed a dynamic that was consistent with slower speeds of less than 1 m/s (100 cm/s), a speed range that includes the speeds at which rats traverse their environment. The next most common category were stationary events (*Yu et al., 2017*; *Farooq and Dragoi, 2019*; *Muessig et al., 2019*) that activated a persistent representation of a single location, while the 'classical' replay events—continuous, extended trajectories through space—made up only about 19% of classified events. Events with slower dynamics tended to persist for approximately the same length of time as events with faster continuous dynamics (~50–100 ms), consistent with a brief activation of a representation of a location or a short trajectory through space. Events with slower dynamics most often represented the animal's current location, similar to continuous events identified in previous work (*Davidson et al., 2009*; *Karlsson and Frank, 2009*); however, slower events were also capable of representing locations far from the animal. Interestingly, *Louie and Wilson, 2001* found that during REM sleep, the longer time scale population activity (on the order of seconds) is highly correlated with firing during times when the animal is running,

resembling the animal's experience. Our results are similar but were observed during pauses in active behavior and on the timescale of SWRs, which last 10s to 100s of milliseconds.

These results challenge the long-held notion that hippocampal replay trajectories necessarily proceed at many times the speed of the animal (*Nádasdy et al., 1999*; *Lee and Wilson, 2002*; *Davidson et al., 2009*). Instead, replay events encompass a much richer variety of event speeds that could promote storage and updating of memories on different spatial and temporal scales, including slower moving representations that activate a spatially compact set of nearby locations and stationary representations of single locations. This is consistent with observations of events that, when decoded in a single time bin, represent a small neighborhood in the environment (*Dupret et al., 2010*). Interestingly, a previous paper (*Farooq and Dragoi, 2019*) reported that these stationary representations were most prevalent in juvenile rats and decreased to 'chance' levels when the rats reached adulthood. Our results strongly suggest that they do not disappear, but are a very common occurrence with adult rats. We believe that the difference in these findings is a result of defining 'chance' levels of occurrence of these events and using a restrictive decoding model which requires the decoded position to be stationary for the entire event. We also note that we chose to include a stationary dynamic to explicitly evaluate the prevalence of stationary events in adult animals, but in principle, a model with just continuous and fragmented dynamics could capture these events as well, as the continuous dynamic includes movement speeds close to 0 m/s.

The differences between our findings and those reported in some previous papers also highlight the importance of identifying the specific question that an analysis seeks to answer. Initial studies of replay (*Lee and Wilson, 2002*; *Foster and Wilson, 2006*; *Davidson et al., 2009*; *Karlsson and Frank, 2009*) focused on the question of whether the observed patterns of spiking seen during replay reflected structured content or 'random' spiking as defined by a comparison to one or more shuffles. These and subsequent studies answered the question 'are replay events the result of random spiking?' with a clear 'no'. These studies did not, however, establish that the significance criteria used to show non-random structure has anything to do with the presence of spatial structure or events that are potentially useful for memory processes.

Our method aims to identify structure within events and to capture their complex dynamics to determine what sort of information replay events could provide in service of memory functions. We precisely define patterns of spatial representation that are, and are not, interpretable in terms of the structure of a given environment and the spatial activity seen during movement. In so doing, we find that most, but not all, events contain spiking that corresponds to a location or a trajectory through the animal's environment.

The different questions addressed by our method compared to previous methods can also explain the apparent differences in findings. Our results showed that 19% of SWRs contained continuous trajectories—meaning that portions of the SWR had spatially coherent trajectories that moved at high velocities—but only 4% were continuous throughout the entire SWR. This, *prima facie*, may seem at odds with previous results, which found between 5% and 45% of SWRs were 'significant', meaning that they were well fit by a line compared to various shuffled versions of the dataset. Furthermore, these studies found trajectories on average move at much higher speeds (20x normal rat movement speed) than we report here. These previous findings were based largely on a single linear fits to the data for each event, while our model allows the trajectory to change speed during the SWR, which can lead to very different estimates of the speed of the event. Many of the events that had continuous dynamics, and would be classified as having high speed trajectories, also exhibited stationary-continuous mixture dynamics for some portion of the event(see for example *Figure 2*). Further, because the linear fit includes the whole SWR event, it is highly sensitive to the definition of the start and end of the SWR event, an often arbitrary definition which varies across research groups. In contrast, our model makes an estimate for each time point and only depends on the previous and future time step, which minimizes the dependence of the event on those boundaries and any possible 'noise' activity within the SWR. This, along with our more permissive threshold for SWRs, allows us to characterize more SWRs than previous studies, rather than exclude them from analysis.

Our results may also explain the seeming conflict between (*Stella et al., 2019*), who reported that hippocampal replay trajectories followed Brownian-like diffusion dynamics, and (*Pfeiffer and Foster, 2015*), who reported that a subset of replay trajectories contained systematic jumps in position. We found that while replay speeds are, on average, distributed log normally, a small subset of the replays include definitive 'jumps' from one position to another. Because our model makes it

possible to identify different dynamics within the same SWR, we are able to identify both types of sequences.

We hypothesize that this large set of events supports the ability to store and update representations of individual places and snippets of experience. Previous findings demonstrated that SWRs with movement- or immobility-related activity engage specific patterns of spiking activity outside the hippocampus (*Jadhav et al., 2016*; *Yu et al., 2017*). Thus, we would expect that the large set of events with slow dynamics would also engage specific patterns of activity outside the hippocampus, perhaps allowing animals to store or update the value of specific locations or short movement trajectories without having to generate full trajectories representing one of multiple ways to arrive at a location (*Yu and Frank, 2015*). Events that contain multiple dynamics (e.g. two continuous representations separated by a brief fragmented representation) might also help bind together experiences that occur in the same environment but at different places, whereas sustained spatially incoherent representations might represent coherent representations in other spatial environments (*Karlsson and Frank, 2009*) or non-spatial experiences. Alternatively, downstream structures may be are attuned to the shifts in dynamics, and that each period of coherent hippocampal dynamics drives its own set of responses elsewhere in the brain. We further note that recent results from a study of activity patterns during movement indicates that individual theta cycles can have different dynamics as well (*Wang et al., 2020*), suggesting that the underlying sequences seen during replay and theta may overlap in their dynamics as well as their content.

Similarly, it is possible that slow and fast dynamics are supported by different subcircuits within the hippocampal formation. *Yamamoto and Tonegawa, 2017* found that blocking medial entorhinal cortex (MEC) output to CA1 resulted in shorter (and therefore slower) replay trajectories and (*Oliva et al., 2018*) showed that layer 2/3 MEC neurons increased their firing 50 ms before the start of longer duration SWRs (>100 ms). This raises the possibility that these slower replays are driven by CA3-CA1 interaction while the faster trajectories are more externally driven by MEC-CA1, and that these different events could also engage different circuits outside of the hippocampus and entorhinal cortex.

One key aspect to understanding how the hippocampus engages with other brain structures is identifying the times when the neural dynamics have changed. The times of SWRs or increases in multiunit activity has provided an important, but coarse, landmark to this change in neural dynamics. Because our method allows for moment-by-moment characterization of the neural dynamic, we suggest that evaluating the start and end of each dynamic may prove a powerful way identify periods of interest. Examination of such periods may advance our understanding of when and why the hippocampus may be representing current position, past experience, or possible future locations, how inputs to the hippocampus influence these events, and how these events may in turn influence downstream structures.

Our identification of the rich and varied dynamics in SWRs depended on several factors. First, we sought to describe as many SWRs as possible, instead of setting a high threshold for event detection. This more permissive approach was also critical for the identification of events that activate specific, immobility-associated locations in previous work (*Yu et al., 2017*). More broadly, there is no evidence that SWR and high population activity events constitute clearly distinct classes of events that can be identified with a single fixed threshold. Instead, the sizes of these events are well described by a unimodal, long-tailed distribution (*Yu et al., 2017*). High thresholds can therefore exclude large numbers of real events and potentially lead to erroneous conclusions.

Second, we used clusterless decoding, which enabled us to take advantage of information from neurons farther away from the tetrodes and not just those that could be cleanly and somewhat subjectively separated (*Chen et al., 2012b*; *Kloosterman et al., 2014*; *Deng et al., 2016*). We note that more fully automated modern spike sorting algorithms (such as MountainSort; *Chung et al., 2017*) in combination with better recording technology could reduce the differences in results when using sorted spikes or clusterless decoding, as these sorting methods help reduce experimenter subjectivity involved in spike sorting and identify larger numbers of putative single neurons.

Third, we built an explicit and flexible model that allowed us to identify not just one dynamic, but multiple dynamics consistent with different speeds of motion. We used these dynamics to describe the SWR events, rather than declaring the events as significant or not.

Fourth, our model avoided using large temporal bins and does not make assumptions about the linear progression of events. Instead, the model allows for movement in any direction allowed by

geometry of the environment, and further, still accounts for the uncertainty of the data. Because the model gives a moment-by-moment estimate of the position and dynamic, this also minimizes the effect of misidentifying the SWR start and end times. Combined with an acausal estimation of latent position, this approach allows us to identify the specific timing of transitions in dynamics at a fine timescale. In addition to our SWR-based analyses, this approach could also be used to decode activity during movement, where we might expect that place-specific activity would express combinations of stationary and continuous dynamics reflecting different speeds of representational movement during each cycle of the theta rhythm. Similarly, because our approach uses the same encoder-decoder approach as the standard decoder, it can be used to investigate sequence dynamics before experience (*Dragoi and Tonegawa, 2013*).

Our model can be viewed as an extension to the multiple temporal models approach of *Johnson et al., 2008*, which used model selection to determine the model with the most appropriate trajectory speed. Our model goes beyond the Johnson model in that it explicitly permits a mixture between the movement speeds, can work for arbitrary track geometries, and uses clusterless decoding. Our model also represents a middle ground between Hidden Markov-style models (*Chen et al., 2012a*; *Chen et al., 2016*; *Linderman et al., 2016*; *Maboudi et al., 2018*), which seek to be environment-agnostic detectors of sequential patterns, and the standard decoder, which typically use arbitrarily chosen bin sizes and restrictive assumptions about the nature of the trajectories. In particular, our approach allows for a variety of dynamics while still yielding spatially interpretable results and makes it possible to use bin sizes of any size (here 2 ms).

Code and algorithms for decoding hippocampal replay have not typically been made accessible or easy to use. This is problematic because it can lead to severe variation or errors in code and parameters, limits reproducibility of results, and slows down scientific progress. Accordingly, we have made code for this model publicly available as a python package at the following URL https://github.com/Eden-Kramer-Lab/replay_trajectory_classification (*Denovellis, 2021a*; copy archived at swh:1:rev:83d84170ae0bdeef65cd123fa83448fcca9cb986). It is easily installable using the pip or conda package installer and contains tutorial Jupyter notebooks in order to facilitate reuse.

State-space models like the one we use here can enable a richer set of analyses of events that take advantage of all of the available data. These approaches can be applied not just to SWRs but to all times, providing a moment-by-moment estimate of the nature of the spatial representation in the hippocampus, including important information about the spatial coherence of that representation. The model can be extended to incorporate other previously experienced environments by training place field models on those environments and including the appropriate movement transition matrices for those environments. It can also be extended to account for task conditions (such as the inbound and outbound conditions in our spatial alternation task) and forward/reverse sequences as in *Deng et al., 2016*. Finally, models can be built to estimate any covariate, including movement direction (*Kay et al., 2020*). We therefore suggest that this approach has the potential to provide a much richer understanding of neural representations throughout the brain.

## Materials and methods

### Simulated data

Encoding data for *Figure 1* and *Figure 1—figure supplement 1* were generated by simulating 15 traversals of a 180 cm linear track. The spiking of 19 neurons was simulated using an inhomogeneous Poisson process where the instantaneous firing rate of each neuron changes according to a Gaussian place field. The Gaussian place fields had a variance of 36 cm and were spaced at 10 cm intervals over the 180 cm track. The decoding data for *Figure 1F* was generated by simulating 20,000 linear traversals of the 180 cm track, each with a unique constant speed, starting at 0.5 cm/s and increasing by 0.5 cm/s up to 10,000 cm/s. Each simulated neuron 'spiked' when the traversal passed through the peak location of its place field.

### Recording locations and techniques

Ten male Long Evans rats (500–700 g, 4–9 months old) were trained on a W-track spatial alternation task. nine rats contributed to previous studies (*Karlsson and Frank, 2009*; *Carr et al., 2012*; *Kay et al., 2016*; *Kay et al., 2020*). Neural activity was recorded in CA1, CA2, CA3, MEC,

Subiculum, and DG depending on the rat. We only used tetrodes located in the CA1, CA2, and CA3 subregions in this study.

## Behavioral task

All animals performed a W-track spatial alternation task, which is described in detail in *Karlsson and Frank, 2009*. In brief, each day, animals alternate between 20 min rest sessions in a rest box and 15 min run sessions on the W-shaped track equipped with a reward well at each arm end. On the W-track, animals are rewarded at the ends of an arm when that arm is the next correct arm in the sequence. Two rules determine the next correct arm. If the animal is in an outer arm, it must next visit the center arm. If the animal is in the center arm, it has to next visit the less recently visited outer arm. Correct performance of these rules result in the visit sequence: center, left, center, right, center, left, etc. Animals were free to choose any arm at any time. Only run recording sessions with at least nine putative hippocampal pyramidal cells that fired at least 100 spikes were included in the analysis.

## Position of the animal and linearization

The animal's 2D position was estimated from digital video (30 Hz) of two infrared diodes placed on the headstage preamplifiers using a semi-automated analysis. In order to decrease the time it takes to run the model, the 2D position of the animal was converted into a 1D position. This is done by first defining a 2D graph representation of the track (herein referred to as the track graph), where edges correspond to segments of the W-track and nodes represent intersection points between those segments. Then, based on the algorithm in *Newson and Krumm, 2009*, we use a Hidden Markov Model (HMM) to assign the position detected in each video frame to the most likely track segment. Using the HMM takes into account the time dependent nature of the data and helps prevents sudden jumps from one track segment to another, which is particularly important near intersections. The observation model of the HMM is Gaussian and it models the likelihood of being on a track segment as the Gaussian distance to that segment with a 5 cm standard deviation. The state transition model is empirically estimated, and changes with each time point to ensure that the Euclidean distance between successive position estimates is similar to the shortest path distance along the graph between successive position estimates. A slight bias of 0.1 is given to the diagonal of the state transition model to encourage staying on the same track segment. The most likely track segment the animal is on is computed using the Viterbi algorithm *Viterbi, 1967*. After finding the track segment that corresponds to each 2D position, the 2D position is projected onto the nearest point of the track segment. This allows us to define a distance from the center well in terms of shortest path length on the track, where 0 cm represents the center well position. The linear distance can then be converted into a linear position by assigning each track segment a position in 1D space. 15 cm gaps were placed between the center arm, left arm, and right arms in 1D space to prevent any smoothing done in the model from influencing neighboring segments inappropriately. The code used for linearization can be found at https://github.com/Eden-Kramer-Lab/loren_frank_data_processing (*Denovellis, 2021b*).

## Sorted spikes, multiunit spikes, and waveform features

To obtain the neural spiking data used for decoding, electrical potentials from rat hippocampus were recorded at 30 kHz, referenced to a tetrode located in the corpus callosum, and then digitally filtered between 600 Hz and 6 kHz. Spiking events were detected as any potential exceeding a 60 $\mu V$ threshold on any one of the four tetrode wires of a tetrode. The electrical potential value on each wire of the tetrode at the time of maximum potential of any of the four wires was used as the waveform feature in the clusterless decoding.

For decoding using sorted spikes, the multiunit events were processed further to assign events to putative single cells. Putative single cells were manually identified based on the clustering of three waveform features within a day: peak amplitude, principal components, and spike width. Only putative hippocampal pyramidal cells—identified based on spike width and average firing rate—were included in the analysis.

## SWR detection

Sharp wave ripples were detected using the same method as in *Kay et al., 2016*. Each CA1 LFP was obtained by downsampling the original 30 kHz electrical potential to 1.5 kHz and bandpass filtering between 0.5 Hz and 400 Hz. This was further bandpass filtered for the ripple band (150–250 Hz), squared, and then summed across tetrodes—forming a single population trace over time. This trace was smoothed with a Gaussian with a 4 ms standard deviation and the square root of this trace was taken to get an estimate of the population ripple band power. Candidate SWR times were found by z-scoring the population power trace of an entire recording session and finding times when the z-score exceeded two standard deviations for a minimum of 15 ms and the speed of the animal was less than 4 cm/s. The SWR times were then extended before and after the threshold crossings to include the time until the population trace returned to the mean value. The code used for ripple detection can be found at https://github.com/Eden-Kramer-Lab/ripple_detection (*Denovellis, 2021b*). We only analyzed SWRs with spikes from at least two tetrodes.

## The model

Let $x_k$ be a continuous latent variable that corresponds to the position represented by the population of cells at time $t_k$ and let $I_k$ be a discrete latent variable that is an indicator for the movement dynamics we wish to characterize: stationary, continuous, and fragmented. The goal of the model is to estimate simultaneously the posterior probability of position and dynamics $p(x_k, I_k \mid O_{1:T})$, where $O_{1:T}$ corresponds to the observed spiking data from time one to time $T$. The observed data can be either spike trains $\Delta N_{1:T}^{(1:C)}$ from $C$ putative cells when decoding with sorted spikes or multiunit spikes $\Delta N_{1:T}^{(1:E)}$ and their associated wave form features $\vec{m}_{k,j}^i$ from each tetrode $E$ when decoding with clusterless spikes, where $i \in 1:E, k \in 1:T, j \in 1:\Delta N_k^i$.

We have previously shown (*Denovellis et al., 2019*) that the posterior probability $p(x_k, I_k \mid O_{1:T})$ can be estimated by applying the following recursive causal filter equation, starting with initial conditions $p(x_0, I_0)$ and iterating to time $T$:

$$p(x_k, I_k \mid O_{1:k}) \propto p(O_k \mid x_k, I_k) \sum_{I_{k-1}} \int p(x_k \mid x_{k-1}, I_k, I_{k-1}) Pr(I_k \mid I_{k-1}) p(x_{k-1}, I_{k-1} \mid O_{1:k-1}) dx_{k-1}$$

and then applying the acausal smoother equation, starting from the last estimate of the casual filter $p(x_T, I_T \mid O_{1:T})$ and recursively iterating backwards to time 1:

$$p(x_k, I_k \mid O_{1:T}) \propto p(x_k, I_k \mid O_{1:k}) \sum_{I_{k+1}} \int \frac{p(x_{k+1} \mid x_k, I_{k+1}, I_k) Pr(I_{k+1} \mid I_k)}{p(x_{k+1}, I_{k+1} \mid O_{1:k})} p(x_{k+1}, I_{k+1} \mid O_{1:T}) dx_{k+1}$$

where:

$$p(x_{k+1}, I_{k+1} \mid O_{1:k}) = \sum_{I_k} \int p(x_{k+1} \mid x_k, I_{k+1}, I_k) Pr(I_{k+1} \mid I_k) p(x_k, I_k \mid O_{1:k}) dx_k$$

Therefore, to specify the model, we have to define or estimate the following quantities:

1. $p(x_0, I_0)$ - the initial conditions
2. $Pr(I_k \mid I_{k-1})$ - the dynamics transition matrix
3. $p(x_k \mid x_{k-1}, I_k, I_{k-1})$ - the dynamics movement model
4. $p(O_k \mid x_k, I_k)$ - the likelihood of the observations

For the initial conditions $p(x_0, I_0)$, we set each dynamic $I_0$ to have uniform $1/3$ probability and each initial latent position to have uniform probability density over all possible positions $\mathcal{U}(\min x, \max x)$, reflecting the fact that we do not have any prior knowledge about which dynamic or position is more likely:

$$p(x_0, I_0) = \frac{1}{3} \mathcal{U}(\min x, \max x)$$

For the dynamics transition matrix $Pr(I_k \mid I_{k-1})$, which defines how likely the dynamic is to change to another dynamic versus persist in the same dynamic, we set it to be:

$$P(I_k \mid I_{k-1}) = \begin{bmatrix} I_K = Stationary & I_K = Continuous & I_K = Fragmented \\ 0.98 & 0.01 & 0.01 \\ 0.01 & 0.98 & 0.01 \\ 0.01 & 0.01 & 0.98 \end{bmatrix} \begin{matrix} I_{k-1} = Stationary \\ I_{k-1} = Continuous \\ I_{k-1} = Fragmented \end{matrix}$$

to encode the prior expectation that each of the dynamics will last the average duration of 100 ms, with a small probability of changing to one of the other dynamics.

For the dynamics movement model $p(x_k \mid x_{k-1}, I_k, I_{k-1})$, which defines how likely the position $x_k$ is to change given the previous position $x_{k-1}$ and current $I_k$ and past dynamics $I_{k-1}$, we set it to be:

$$P(x_k \mid x_{k-1}, I_k, I_{k-1}) = \begin{bmatrix} I_K = Stationary & I_K = Continuous & I_K = Fragmented \\ \delta(x_{k-1}) & \mathcal{N}(x_{k-1}, 6.0) & \mathcal{U}(\min x, \max x) \\ \delta(x_{k-1}) & \mathcal{N}(x_{k-1}, 6.0) & \mathcal{U}(\min x, \max x) \\ \mathcal{U}(\min x, \max x) & \mathcal{U}(\min x, \max x) & \mathcal{U}(\min x, \max x) \end{bmatrix} \begin{matrix} I_{k-1} = Stationary \\ I_{k-1} = Continuous \\ I_{k-1} = Fragmented \end{matrix}$$

where $\delta(x_{k-1})$ is an identity transition matrix where position cannot change from the previous time step, $\mathcal{N}(x_{k-1}, 6.0)$ is a random walk from the previous position with variance 6.0, and $\mathcal{U}(\min x, \max x)$ is a uniform transition that allows transitions to any possible position. As discussed in the Results, this means that when persisting in the same dynamic, the stationary, continuous, and fragmented dynamics are defined by the identity transition, the random walk, and the uniform transition, respectively. When transitioning to or from the fragmented dynamic, we assume we do not have any information about the position, so the transition is uniform. Finally, when the transition is from the stationary to continuous, we assume the position is spatially close where it was previously, so we use a random walk. When the transition is from continuous to stationary, we assume that the position is no longer changing, so we use the identity transition.

Lastly, we evaluate the likelihood of the observations $p(O_k \mid x_k, I_k)$ based on an encoding model fit during the encoding period. We assume the likelihood is the same for each dynamic $I_k$, so we only need to evaluate $p(O_k \mid x_k)$. It has been shown (*Zhang et al., 1998*; *Brown et al., 1998*) that the Poisson likelihood with sorted spikes can be computed as:

$$p(O_k \mid x_k) = p(\Delta N_k^{(1:C)} \mid x_k) \propto \prod_{i=1}^{C} [\lambda_i(t_k \mid x_k)\Delta_k]^{N_{t_k}^i} \exp[-\lambda_i(t_k \mid x_k)\Delta_k]$$

where $N_{t_k}^i$ represents a spike at time $t_k$ from cell $i$, $\Delta_k$ is the time bin size, and $\lambda_i(t_k \mid x_k)$ is the instantaneous firing rate of cell $i$ given position $x_k$. $\lambda_i(t_k \mid x_k)$ is the 'place field' of the cell, which can be estimated by fitting a generalized linear model to each cell's spiking during the encoding period.

Likewise, it has been shown (*Chen et al., 2012b*; *Kloosterman et al., 2014*; *Deng et al., 2016*) that the clusterless likelihood can be computed as:

$$p(O_k \mid x_k) = p(\Delta N_k^{(1:E)}, \{\vec{m}_{k,j}^i\}_{j=1:\Delta N_k^i}^{i=1:E} \mid x_k) \propto \prod_{i=1}^{E} \prod_{j=1}^{\Delta N_k^i} [\lambda_i(t_k, \vec{m}_{k,j}^i \mid x_k)\Delta_k] \exp[-\Lambda_i(t_k \mid x_k)\Delta_k]$$

where $\lambda_i(t_k, \vec{m}_{k,j}^i \mid x_k)$ is now a generalized firing rate that depends on an associated wave form features $\vec{m}$ and $\Lambda_i(t_k \mid x_k)$ is a marginal firing rate that is equivalent to a place field estimated on multiunit spikes. Both of these rates can be defined as:

$$\lambda_i(t_k, \vec{m}_{k,j}^i \mid x_k) = \mu_i \frac{p_i(x_k, \vec{m}_{k,j}^i)}{\pi(x)}$$

and

$$\Lambda_i(t_k \mid x_k) = \mu_i \frac{p_i(x_k)}{\pi(x)}$$

where $\mu_i$ is the mean firing rate for tetrode $i$, $\pi(x)$ is the smoothed spatial occupancy of the animal on the track, $p_i(x_k)$ is the smoothed spatial occupancy only at times when spikes occur in tetrode $i$, and $p_i(x_k, \vec{m}_{k,j}^i)$ is the smoothed occupancy in the space of both space and waveform features. $\pi(x)$,

$p_i(x_k, \vec{m}_{k,j}^i)$, $p_i(x_k)$ can all be estimated by training a kernel density estimator on each tetrode's spike waveform features and corresponding position during the encoding period.

## Encoding - sorted spikes

In order to encode how each cell's spiking activity relates to position (the place field), we fit a generalized linear model (GLM) with a Poisson response distribution to each cell's spiking activity during the encoding period, which we define as all movement times (time periods when the running speed is greater than 4 cm/s). We estimate the parameters $\beta$, which consist of $\beta_0$, the baseline firing rate over time, and $\beta_i$, weights for third degree B-spline basis functions $f_i$ over position (or tensor products of the B-splines when position is two dimensional). B-spline basis functions are used because place field firing activity is assumed to vary smoothly over position and this prior knowledge can be exploited to reduce the total number of model parameters needed. Each basis function is spaced every 5 cm over the range of the position and zero constrained so that the change encoded by the parameters is relative to the baseline firing rate. We use a log link function to convert the linear combination of parameters to an instantaneous firing rate over time $\lambda(t_k)$ to ensure the rate is always positive.

$$log(\lambda(t_k)) = \beta_0 + \sum_i f_i(x_k)\beta_i$$

A small L2 penalization term $-\lambda\|\beta_i\|_2^2$ is used to prevent model fitting instability when spiking activity is very low. We set this to 0.5 for all cells. Fitting is done by maximizing the penalized likelihood using a Newton-Raphson algorithm. The code used to fit the GLMs is available at https://github.com/Eden-Kramer-Lab/regularized_glm; *Denovellis, 2021b*.

## Encoding - clusterless

In order to relate each tetrode's unsorted spiking activity and waveform features to position during the encoding period, which we define as all movement times (time periods when the running speed is greater than 4 cm/s), we used kernel density estimation (KDE) to estimate the following distributions: $\pi(x)$, $p_i(x_k)$, and $p_i(x_k, \vec{m}_k)$. We used KDEs of the form:

$$kde(y) = \frac{1}{Nh_1 \cdots h_D} \sum_{i=1}^{N} \prod_{d=1}^{D} K_d\left(\frac{y_d - y_{i,d}}{h_d}\right)$$

where $y$ is the data with $D$ dimensions, $K$ is a one dimensional Gaussian kernel with bandwidth $h_d$, $y_{i,d}$ is the data observed during movement, $N$ is the number of observations during the movement period. For $\pi(x)$, $y_{i,d}$ is all positions observed during movement. For $p_i(x_k)$, $y_{i,d}$ is all positions at the time of multiunit spikes during movement. For $p_i(x_k, \vec{m}_{k,j}^i)$, $y_{i,d}$ is all positions at the time of multiunit spikes and their associated waveform features during movement. We choose the bandwidth $h_d$ to be 6.0 cm for any position dimension and 24.0 $\mu V$ for any spike amplitude dimension, because these parameters were found to minimize decoding error in *Kloosterman et al., 2014*. The mean firing rate $\mu_k$ was also estimated during movement.

## Decoding

In order to decode $p(x_k, I_k \mid O_{1:T})$, we used a grid-based approximation of the latent position $x_k$ that respected the geometry of the track. For 1D linearized positions, we discretized the position space based on the same track graph we used for linearization by finding bins less than or equal to the chosen position bin size for each edge of the graph. For 2D positions, we discretized the position space by binning 2D positions occupied by the animal with equal sized bins of the chosen position bin size, followed by morphological opening to get rid of any holes smaller than the bin size. This was done using Scipy image processing tools (*SciPy 1.0 Contributors et al., 2020*). This grid based approximation allows us to use Riemann sums to approximate the integrals in the causal filter and acausal smoother equations. We chose a position grid with bins of 3 cm in width (and height if the model was computed with 2D positions) in order to get good resolution for the random walk transition matrices (which had 6 cm variance) as well as for the clusterless and sorted spikes decoding (which have 6 cm bandwidth for the KDE position dimensions and 5 cm spline knots for the GLM).

For sorted spikes decoding, we evaluated the place field $\lambda_i(t_k \mid x_k)$ on the midpoint of these bins. Likewise, for clusterless decoding, we evaluated the spike amplitudes observed during the decoding period by evaluating the KDE for $p_i(x_k, \vec{m}_{k,j}^i)$ for the midpoint of these bins.

We also used these grid bins in combination with the track graph in order to construct the appropriate 1D random walk transition matrices that respected the track geometry. To do this, we inserted the bin centers as nodes in the track graph, and then computed the shortest path distance between all pairs of position bin centers. We then evaluated a zero mean Gaussian with a variance of 6 cm on these distances to get the appropriate transition probability of moving from bin to bin.

Finally, for our SWR analysis of $p(x_k, I_k \mid O_{1:T})$, we decoded each immobility time period (times when the animal's speed was less than 4 cm / s) in 2 ms time bins and extracted the SWR times.

## Probability of the dynamics

The probability of the dynamics is a quantity that indicates how much that dynamic is contributing to the posterior at a given time. We estimate the probability of the dynamics by integrating out position from the joint posterior:

$$Pr(I_k \mid O_{1:T}) = \int p(x_k, I_k \mid O_{1:T}) dx_k$$

As in other calculations, the integral is approximated with a Riemann sum.

## Posterior probability of position

The posterior probability of position is a quantity that indicates the most probable 'mental' positions of the animal based on the data. We estimate it by marginalizing the joint probability over the dynamics.

$$p(x_k \mid O_{1:T}) = \sum_{I_k} p(x_k, I_k \mid O_{1:T})$$

## Classification of dynamics

We used a threshold of 0.80 to classify the probability of each state $Pr(I_k \mid O_{1:T})$ into five categories. Time periods during sharp wave ripples are labeled as stationary, continuous, or fragmented when the probability of each individual state is above 0.80, regardless of duration. Time periods are labeled as stationary-continuous-mixture or fragmented-continuous-mixture when the sum of stationary and continuous or fragmented and continuous are above 0.80, respectively. Time periods where none of these criterion are met are considered unclassified.

## Highest posterior density

The 95% highest posterior density (HPD) is a measurement of the spread of the posterior probability and is defined as the region of the posterior that contains the top 95% of values of the posterior probability (*Casella and Berger, 2001*). By using only the top values, this measurement of spread is not influenced by multimodal distributions (whereas an alternative measure like the quantiles of the distribution would be). In this manuscript, we use the HPD region size—the total area of the track covered by the 95% HPD region—to evaluate the uncertainty of the posterior probability of position.

In order to calculate the HPD region size, we first calculate the 95% HPD region by determining the maximum threshold value $h$ that fulfills the follow equality:

$$\int_{\{x:p(x_k \mid O_{1:T})>h\}} p(x_k \mid O_{1:T}) dx = 0.95$$

The 95% HPD region is the set of position bins with posterior values greater than the threshold $\{x : p(x_k \mid O_{1:T}) > h\}$. The HPD region size is calculated by taking the integral of the members of this set:

$$\int_{\{x:p(x_k \mid O_{1:T})>h\}} \mathbb{1}\, dx$$

which we approximate with a Riemann sum.

## Shuffle analysis of the effect of place encoding on classification of dynamics and HPD region size

In order to confirm that the model classification of dynamics depended on hippocampal place specific encoding, we resampled the position during movement with replacement, but preserved the spike times and spike waveform amplitudes. We fit the clusterless encoding model on the resampled data and then decoded the immobility periods. Like with the non-resampled data, we then extracted the SWR times and determined their classification based on our classification scheme. We repeated this shuffle analysis 50 times and compared this distribution to the real data for two recording sessions on two different animals (animal Bond, day 3, epoch two and animal Remy, day 35, epoch 2).

We also performed another shuffle that preserved more of the local correlation structure of spiking by shuffling the order of the well-to-well runs and then circularly shuffling the position within the runs. A well-to-well run starts when the animal moves more than 5 cm from a well and ends when the animal moves back within 5 cm of a well. This largely reflects runs from one well to another (e.g. from the center well to the left well), but for a small number of well-to-well runs, the animal did turn back to the same well (e.g. went back to the center well after leaving the center well). As above, after permuting the order of the well-to-well runs and circularly shuffling the position, we fit the clusterless encoding model and decoded SWRs. We repeated this shuffle analysis 50 times and compared this distribution to the real data for two recording sessions on two different animals (animal Bond, day 3, epoch two and animal Remy, day 35, epoch 2).

## Distance from animal

The distance of the decoded position from the animal's is defined as the shortest path distance along the track graph between the animal's 2D position projected on to the track graph (see Linearization) and the MAP estimate of the posterior probability of position, $*arg\,max_{x_k} p(x_k \mid O_{1:T})$, which is the center of the position bin with the greatest posterior value. The shortest path is found using Dijkstra's algorithm *Dijkstra, 1959* as implemented in NetworkX (*Hagberg et al., 2008*).

## Estimation of speed

In order to estimate the speed over time, we first found the MAP estimate of the posterior probability of position, $*arg\,max_{x_k} p(x_k \mid O_{1:T})$, which is the center of the position bin with the greatest posterior value. We then computed the first derivative using the Python library Numpy's gradient function, which computes the difference in the forward and backward time direction and then averages them for all points except the boundaries. We then smoothed this quantity with a small Gaussian (2.5 ms standard deviation) and then averaged this for the classified time points. Because our position bin size of 3 cm makes it hard to distinguish between slower speeds in a 2 ms time step, we only analyzed classifications longer than 20 ms.

## Non-local stationary position

The non-local stationary position is defined as replay distances at least 30 cm from the animal's position during a time period classified as stationary.

## Identifying events of high multiunit activity

We identified times of high multiunit activity when the animal was immobile as a control analysis. Our approach was similar to *Davidson et al., 2009*. High multiunit periods were identified as times when the z-scored multiunit population spiking activity was greater than two standard deviations for at least 15 ms and the animal was moving at speeds less than 4 cm/s. The code used for detection of high multiunit time periods can be found at https://github.com/Eden-Kramer-Lab/ripple_detection.

## Standard 'Bayesian' decoding and significance estimation

For the standard decoder analysis, we first fit a clusterless encoding model in the same manner as described in the subsection Encoding - Clusterless. We then decoded the posterior probability of each position, using 20 ms time bins, using the clusterless likelihood equation as described in the subsection The Model and assuming a uniform prior probability. Finally, to get an estimate of speed of the replay, we used either the Radon transform method, the linear regression method, or the MAP estimate method.

For the Radon transform method, we densely sampled potential line trajectories and picked the line that maximized the probability along that line. The sum of the probability along the line is called the Radon score. We picked the maximum Radon score on lines fit on the combination of center arm and left arm posterior versus center arm and right arm posteriors. p-Values were computed by circularly shuffling the posterior probabilities 1000 times and comparing the real data vs. the shuffled distribution.

For the linear regression method, we drew 1000 position samples from each time bin according to the probability of position in that time bin. A linear regression was then fit to the sampled positions using time as a covariate. Fits were made separately for the posterior probabilities corresponding to the center arm with the left arm and the center arm with the right arm. Quality of the fit was assessed using the R-squared and the best fit between the arms was selected based on this metric.

For the MAP estimate, we pick the position bin with the maximum value for each time bin, which corresponds to the most probable position.

## Software and code availability

Python code used for analysis and generating figures in the paper is available at: https://github.com/Eden-Kramer-Lab/replay_trajectory_paper (*Frank, 2021* copy archived at swh:1:rev:630d4e32f343e86a4921d0773f1f5c5adf90553f). Code for the decoder is available in a separate software repository to facilitate code reuse at: https://github.com/Eden-Kramer-Lab/replay_trajectory_classification (*Denovellis, 2021a*; copy archived at swh:1:rev:83d84170ae0bdeef65cd123fa83448fc-ca9cb986) (doi: 10.5281/zenodo.3713412). Both code bases rely on the following python packages: Numpy (*van der Walt et al., 2011*), Numba (*Lam et al., 2015*), Matplotlib (*Hunter, 2007*), xarray (*Hoyer and Hamman, 2017*), NetworkX (*Hagberg et al., 2008*), Pandas (*McKinney, 2010*), and Seaborn (*Waskom, 2021*). All code is open-source and licensed under the MIT Software License. Decoder code can be easily installed as a python package with all requisite dependencies using pip or conda. See software repositories for specific details.

## Acknowledgements

We thank Margaret Carr Larkin and Mattias P Karlsson for use of their openly available datasets. We also thank Abhilasha Joshi for feedback on the manuscript. This work was supported by the Simons Foundation for the Global Brain Grants 542971 (UTE) and 542981 (LMF) and the Howard Hughes Medical Institute (LMF).

## Additional information

### Funding

| Funder | Grant reference number | Author |
| --- | --- | --- |
| Simons Foundation | 542971 | Uri T Eden |
| Simons Foundation | 542981 | Eric L Denovellis<br>Anna K Gillespie<br>Michael E Coulter<br>Marielena Sosa<br>Jason E Chung<br>Loren M Frank |
| Howard Hughes Medical Institute | | Eric L Denovellis<br>Loren M Frank |

The funders had no role in study design, data collection and interpretation, or the decision to submit the work for publication.

## Author contributions
Eric L Denovellis, Conceptualization, Data curation, Software, Formal analysis, Visualization, Writing - original draft, Writing - review and editing; Anna K Gillespie, Conceptualization, Writing - review and editing; Michael E Coulter, Writing - review and editing; Marielena Sosa, Jason E Chung, Data curation, Writing - review and editing; Uri T Eden, Resources, Formal analysis, Supervision, Funding acquisition, Methodology, Writing - review and editing; Loren M Frank, Conceptualization, Resources, Supervision, Funding acquisition, Writing - original draft, Writing - review and editing

## Author ORCIDs
Eric L Denovellis ⓘ https://orcid.org/0000-0003-4606-087X
Anna K Gillespie ⓘ https://orcid.org/0000-0003-0980-2408
Marielena Sosa ⓘ https://orcid.org/0000-0003-0762-1128
Loren M Frank ⓘ https://orcid.org/0000-0002-1752-5677

## Ethics
Animal experimentation: All experiments were conducted in accordance with University of California San Francisco Institutional Animal Care and Use Committee and US National Institutes of Health guidelines. The protocol was approved by the Institutional Animal Care and Use Committee, approval number AN174991-03G. All surgical procedures were performed under anesthesia and every effort was made to minimize suffering.

## Decision letter and Author response
Decision letter https://doi.org/10.7554/eLife.64505.sa1
Author response https://doi.org/10.7554/eLife.64505.sa2

# Additional files
## Supplementary files
• Transparent reporting form

## Data availability
All data from this paper has been deposited in a dryad repository. Some of this data was previously deposited at the CRCNS data sharing website as dataset hc-6, but for reproducibility purposes we have included it here.

The following dataset was generated:

| Author(s) | Year | Dataset title | Dataset URL | Database and Identifier |
|---|---|---|---|---|
| Denovellis EL, Gillespie AK, Coulter ME, Sosa M, Chung JE, Eden UT, Frank LM | 2021 | Data from: Hippocampal replay of experience at real-world speeds | https://doi.org/10.7272/Q61N7ZC3 | Dryad Digital Repository, 10.7272/Q61N7ZC3 |

The following previously published dataset was used:

| Author(s) | Year | Dataset title | Dataset URL | Database and Identifier |
|---|---|---|---|---|
| Karlsson MP, Carr M, Frank LM | 2015 | Simultaneous extracellular recordings from hippocampal areas CA1 and CA3 (or MEC and CA1) from rats performing an alternation task in two W-shaped tracks that | http://dx.doi.org/10.6080/K0NK3BZJ | Collaborative Research in Computational Neuroscience, 10.6080/K0NK3BZJ |

are geometrically identically but
visually distinct

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
