## [Decision Letter]

**Acceptance summary:**

This paper presents a new framework to decode neuronal activity in the hippocampus during ripple events. This method reveals that most ripple events contain spatially interpretable content that often progresses at timescales slower than previously reported and that appears to better match previous wake activity at real time. These findings challenge the classical view of replay as primarily time-compressed representation of previous wake activity and provide a fuller picture of the repertoire of possible content for ripple events during rest.

**Decision letter after peer review:**

Thank you for submitting your article "Hippocampal replay of experience at real-world speeds" for consideration by *eLife*. Your article has been reviewed by 3 peer reviewers, and the evaluation has been overseen by a Reviewing Editor and Timothy Behrens as the Senior Editor. The following individuals involved in review of your submission have agreed to reveal their identity: Yunzhe Liu (Reviewer #1); H Freyja Ólafsdóttir (Reviewer #2).

The reviewers have discussed the reviews with one another and the Reviewing Editor has drafted this decision to help you prepare a revised submission.

Summary:

In the present study, the authors report that the proportion of candidate replay events in the hippocampus showing spatially interpretable content is much higher than previously described. In addition, these events often progress at slow ("real-world") timescales while it is generally assumed that hippocampal replay is compressed in time. Specifically, the authors have used a state space model that captures the spatial content and temporal evolution of replay. This model is based on few assumptions and is aimed to be unbiased – as it is based on "clusterless" extracellular spikes. While the three reviewers are generally enthusiastic about the work and agree that this study is potentially suitable for *eLife*, they have raised several concerns that should be addressed to warrant publication. The main concerns are summarized below, please refer to the detailed reviewer's comments for more information.

Essential revisions:

1. The authors should further test their method on exploration data to validate the approach, especially since non place cells may potentially bias the decoding of animal's speed in clusterless data (see reviewer #3 point #1). This analysis of hippocampal activity during run can be used to identify the encoded speed in within theta sequences (reviewer #1, point #2).

2. The authors have to improve controls, especially their shuffling procedures. Shuffling the spike-location relationship is too permissive. See reviewer #2, point #1 and reviewer #3, point #9.

3. The authors should use more stringent thresholds on speed and theta power to rule out the possibility that some of the events are not replays but represent current position. See reviewer #2, points #2-3.

4. The classification of candidate replay events should be improved (especially events that may be partially "unclassified") and the authors should provide a more detailed comparison of the replay events detected with their method and with previously described methods (reviewer #3, points #2-3)

5. The authors should provide descriptive statistics demonstrating that spiking properties during candidate replay events (e.g. average rate per tetrode) are similar between sorted and clusterless data (reviewer #3 point #6-7). Only events in which spikes are detected on multiple tetrodes should be included, to prevent decoding from the firing of a single cell. Finally, spikes of putative inhibitory neurons should be discarded (reviewer #3, point #7).

6. The authors have used neuronal data collected from the different subcircuits of the hippocampus (CA1-3). It should be demonstrated that the main results hold if the analysis is restricted to CA1, on which most of previously published studies are based, especially since it can affect the nature of replay events. Is there a relationship between the category (e.g. continuous or stationary) of a given replay event and the proportion of neurons from the different substructures that fired during that event? Also, discussing the seemingly low proportion of continuous events (if the same proportion is observed when only including CA1 neurons). See reviewer #2, point #4 and reviewer #3, points #4-5.

7. The title of the manuscript is a bit misleading as the reported replayed speeds are often even slower than "real-world speed" and should therefore be changed. See reviewer #1, point #1 and reviewer #3, point #8.

8. The authors should further discuss the implication of their findings. See reviewer #1, point #3 and reviewer #2, point #4.

9. Please note that reviewer #3 has also raised a couple of minor concerns that should be addressed.

*Reviewer #1:*

Denovellis et al. have developed a state space model approach for unbiased replay detection. As a result, they are able to demonstrate that most of the candidate replay events (i.e., sharp wave ripple events) contains spatial coherent (stationary or continuous) content, and they progress at a relative lower speed (< 1m/s) than previously believed.

I like this paper a lot. The writing is very clear, and the analysis is thorough. The proposed method is likely to be very impactful and widely adopted in the replay community.

*Reviewer #2:*

Denovel and colleagues develop and apply a state space model to decode position representation during hippocampal SWR events. The main advantage of this model is that is makes fewer assumptions of the dynamics of SWR events allowing it capturing varying and mixed movement dynamics. Through the application of this model the authors are able to decode coherent spatial content from the vast majority (~90%) of SWR events – a much, much higher proportion than previous papers on the topic have reported – and they show that, contrary to previous work, the most common type of movement dynamic during SWR events is one that's characterised by a stationary/slow movement speeds.

I believe the work by Denove and colleagues represents a significant advance in our understanding of SWR events. Moreover, the state space method Denove et al. described also represents useful analytical tool which could potentially be useful to many neuroscientists in the field. I would also like to commend the authors for their clear and detailed explanation of the model.

I do however have some concerns about the robustness of their SWR event method and some of their control analyses. Below I explain these in more details and suggest ways to address these concerns. All suggestions are implementable and would, I believe, significantly strengthen the authors' conclusions.

1. To control for spurious SWR events (containing spatially coherent content), the authors shuffle the position-spike relationship. Although this type of control analysis is commonly used, it can be overly permissive, particularly if researchers are working with small-to-medium samples sizes. This is because for smaller samples sizes it is unlikely that all positions are represented equally by the cell population (which is an assumption of the control analysis). However, given the authors are using clusterless decoding, a cell ID shuffle is not possible. Thus to address this potential caveat, it would be useful if the authors show position representation in their cell population on the W-maze? Moreover, if the position representation is not equal throughout the environment the authors need to account for this in their position resampling with replacement control analysis.

2. The authors use a speed threshold of 4cm/s to exclude SWR events that may occur during movement periods. Could the authors use a more stringent threshold – e.g. 1cm/sec or 0cm/sec? Although 4cm/sec is low I believe it is important to rule out the possibility that some of their detected spatially coherent SWR events don't merely reflect the movement of the animal during the event. This is particularly important given their finding that the most common type of replay event is one that is characterised by slow movement speeds – akin to those seen during actual behaviour.

3. Related to this, the authors could also use low theta power as an additional quality criteria to weed out events that may reflect (slow) movement periods.

4. One of the main findings of the paper is that SWR events can consist of different types of movement dynamics (stationary, continuous, fragmented) or a mixture of dynamics. This is indeed an interesting and novel finding, but it would be useful if the authors could elaborate on the theoretical implications of different type of replay events. For example, do the authors think SWR events belonging to different movement dynamic categories are supported by different sub-circuits within the hippocampus? For example, some work has shown SWR events preceded by EC activity are longer in duration Oliva et al. (2016).

*Reviewer #3:*

The authors of this study investigated the prevalence and range of representational dynamics associated with replay activity in the rat hippocampus during periods of rest following a spatial alternation task in a W-shape maze. Using a state space model that captures the spatial content and temporal evolution of replay during sharp-wave ripple activity, the authors report that most ripple events contain spatially interpretable content that often progresses at timescales slower than previously reported and that appears to better match previous wake activity at real time. The reported findings challenge the classical view of replay as primarily time-compressed representation of previous wake activity and build on previous work on stationary events and reduced proportion of significant trajectory replay events to provide a fuller picture of the repertoire of possible content for ripple events during rest. While the results are intriguing, a number of conceptual and technical shortcomings prevent a more enthusiastic appreciation of this work in its current state, as detailed below.

1. The main Bayesian model needs to be tested first during run on track, where the prior is being built. The authors need to compute the error of the model during run in terms of the difference between the decoded position during run and the actual animal trajectory. If, for any reason, the model appears stationary during run at any trial/lap, then the authors will need to take that into account when claiming stationarity during rest. In particular, clusterless decoding may make use of spikes that are not spatially well modulated and that would appear to support the stationary events during rest.

2. Please use more stringent classification of the events. As the current relaxed criterion for classifying events only requires a portion of the event to exhibit that behavior, what proportion of events classified as 'stationary' or 'continuous' significantly pass traditional, rigorous event-matched shuffles? What proportion of extended stationary events, >100 ms for instance, have a line-fit replay score above the 95th percentile of event-matched shuffles (circular shifting of posterior decoded probabilities) and a trajectory length shorter than a certain value, e.g. 6 cm? Similarly, what proportion of continuous trajectories pass their respective event-matched shuffles? What proportion of stationary events are entirely stationary? The authors need to quantify all of the above and present them in the Results.

3. One significant conclusion of this paper is that most population events during awake rest are stationary-continuous mixtures or stationary as opposed to continuous (replay) events. However, given that an event is classified as stationary if part of the event is stationary (probability >0.8), there is a possibility that the rest of the event is unclassified (no clear structure). In plots 3G, 5A-F it is important to include details about the unclassified ripples and the unclassified parts of the ripple which is otherwise classified. For instance, in 5A, the proportion of ripples with parts that are unclassified needs to be mentioned. This is important because parts of the event might be stationary, and parts might be noise/unclassified. Previous studies have characterized events as stationary only if they were stationary throughout the event and had excess stationarity than the 95th percentile of an event-based shuffle. How many of the events classified as stationary pass this significance criterion? This relaxed classification might otherwise lead to over-classification of certain event types.

4. The reported results are based on combined recordings from 3 hippocampal areas: CA1, CA2, and CA3 (conform Methods). These areas have different cell types, some of which respond to animal stationarity rather than movement. This makes classification and comparison of all event types between this study and the previous studies reporting on CA1 place cells only which were active on simpler tasks on linear tracks difficult. The authors should restrict their analysis to tetrodes located exclusively in CA1 when computing proportions of event types and only then compare their results to previous studies reporting on significant replay and stationary events. To make this comparison even more interpretable, similar significance tests against shuffles and use of clustered data (available for 9 out of the 10 animals that were previously published) should be performed. If the presented proportions of events persist under these conditions, the impact of the findings will be significantly increased.

5. There is discrepancy in proportion of replay events reported here compared to previous studies. Continuous events are most similar to traditionally detected replay sequences, however, those sequences are unidirectional (move from one side of the track to the other to be significant), while these events can move in both directions (model of movement dynamic is symmetric for continuous). The authors should investigate and explain why the proportions of continuous events are less than those of traditionally detected replay sequence events with event-matched shuffles (10-50% in other studies as stated by the authors versus ~5% only continuous).

6. The properties of each event type need to be compared to rule out other reasons for their distinct spatial and temporal dynamics. These properties to be compared between classes of event types (e.g. stationary, continuous, etc.) include: number of tetrodes active in the event, ripple power, total number of spikes in the event, spikes per bin, event duration, proportion of spikes under 6 ms ISI for each tetrode in each event type, proportion of the event duration it lies in that classification. Could you also confirm that population events included for clusterless decoding are similar in properties to those obtained from clustered data by comparing their properties (duration and z-scored population firing rates across the event)?

7. There is a possibility that one neuron dominates a population event and, therefore, stationary events are simply single units firing continuously. For instance, Figure 1A for the simulation shows just that, one simulated neuron fires and a stationary event is predicted by the algorithm. As clusterless decoding is used for the main analyses, can only events consisting of multiunit activity from at least 5 tetrodes be included in the analysis? Will the main classes of events be replicated under these conditions? Related to this, interneurons (putative inhibitory neurons) should be removed from the clusterless decoding based on waveform shape as there is a possibility that inclusion of interneurons in the clustering decoding analysis biases results towards detection of stationary events (there might be a hint of this from examples presented in Figure S4 where good stationary events seem to have higher spiking per tetrode compared to continuous events). The event classes should be recomputed and compared with the original classes.

8. The current title is misleading. There does not appear to be clear evidence of replay at real-world speeds. First, during each run lap, the animal trajectory changes unidirectionally at an estimated speed of 25-35 cm/s. For replay to occur at real time speed, the trajectory decoded during rest should fully unfold at the same speed over 3-4 s for a 1m-long track. This is never reported here, partly because the events are shorter. The stationary events decode individual locations, but there is no indication that they evolve in succession (successive locations at successive times) to represent trajectories at real time speeds. Second, neurons fire in time-compressed theta sequences during run, so neuronal activity is naturally compressed even before replay. Compressed replay essentially already represents space at roughly the same speed as during run (theta sequences).

9. Bursting of neurons or other individual neuronal properties might lead to classification of successive bins as spatially coherent and/or stationary. In order to control for that possibility, in clustered decoding researchers typically generate shuffled datasets with conserved place map properties by circularly shifting individual place maps. The shuffles generated in this paper randomize the position to spike relationship destroying all single-cell properties. Can shuffles with conserved tetrode multiunit activity properties be generated instead to test this hypothesis? E.g. circularly shift by the same value all spikes from the same tetrode.

[Editors' note: further revisions were suggested prior to acceptance, as described below.]

Thank you for resubmitting your work entitled "Hippocampal replay of experience at real-world speeds" for further consideration by *eLife*. Your revised article has been evaluated by Timothy Behrens (Senior Editor) and a Reviewing Editor.

The manuscript has been improved but there are some remaining issues that need to be addressed, as outlined below:

The most critical aspect is about the decoding. During wake, theta sequences should strongly impact the accuracy of the decoding with 2m time bins, but it is unclear why this is not the case. The study should also provide some comparison of decoding accuracy with 2ms and 20ms and provide further details regarding how exactly events are eventually classified (see reviewer #2, points 9-11).

The study should also include examples showing the differences and similarities in decoding with the clusterless and more classical approaches (see reviewer #2, point #1).

Some other concerns were raised, mostly requesting clarification in the methods and the discussion of the results. Please see detailed review below (reviewer #2, points #2-8).

*Reviewer #2:*

I am satisfied with the changes the authors have made in their manuscript and the updated manuscript addresses all of my concerns.

I think the findings are exciting and will contribute to theoretical development in the field. I particularly like that the authors have made the code for their state space model available – this may prove useful for promoting methodological consistency and transparency in the replay field.

*Reviewer #3:*

In the revised manuscript, the authors have partially addressed the previous concerns. The manuscript has improved, and the findings remain potentially interesting. Below, please find additional comments aimed at improving the accuracy and readability of the manuscript:

1. Due to the various differences from earlier methods (clusterless decoding, state space encoding model, bin size, event detection criteria and quantification of dynamic type: radon transform vs current method), it is difficult to pinpoint which of these differences is leading to the authors' main conclusions. For the general readership, it will be extremely useful to show actual examples of the same decoded spike trains using standard Bayesian decoding (with clustered units) and using state space models. This could reveal any significant differences in the decoded locations which the authors claim are revealed by this method (pages 3 and 4). Since the authors claim there is a substantial difference between the two methods, differences in the inferred dynamics from the two methods should be visible by eye and many such examples should be available.

2. Is the higher preponderance of stationary/slow events caused by the authors setting of the beginning and ends of events as the start and ends of ripples, rather than using the multiunit population activity (which can span multiple ripples). Can the authors rerun the analysis with multiunit population activity to ensure that the results are not different from previous studies mostly due to how events are detected? Even if the results are due to that, it will be useful to report that to the reader as it will not diminish the impact of these findings (the authors already allude to the fact that the cut off is arbitrary) but will help them understand the reason for the differences between the studies.

3. Can the authors include a histogram of event duration versus proportion of such events for each of the dynamic so that the reader can better judge what is the likelihood of each event type by duration?

4. Although the fragmented dynamic would indeed identify discontinuities in decoded locations, how exactly is it better than a non-parametric shuffle? What if the fragmented dynamic has more false-positives or false-negatives?

5. Are the discovered dynamics restricted to the replay of the animal experience or can they be present before the experience? The authors should discuss this possibility.

6. How is stationarity a real-world 'speed'? Isn't it a lack of movement? Were only the spikes emitted while the animal was moving used for the encoding model and not those while it was resting? How are then the stationary events replays of that movement? What was the average velocity of the animal during movement? And of the stationary dynamic?

7. References cited and conclusions drawn from papers are not accurate in some instances, please verify and correct. Several cases are listed below, but the list is larger. Louie and Wilson, Neuron 2001 showed that replay of experience can occur at real-time speed during REM sleep, which should be cited. On lines 67-68 the authors say: "events that represent a single location are only seen in young animals (Stella et al., 2019)". That study did not investigate young animals, the authors probably meant to refer to (Farooq and Dragoi, 2019), which should be cited. There is a difference between what previous studies have claimed, and what the authors attribute to those studies. For instance, previous studies observed low proportions of sequential trajectories, but they did not observe low proportions of spatially coherent time-bins/cofiring, which are otherwise well-reported and have been observed many times.

8. How do the authors know that a probability of 0.8 or above can be inferred from downstream neurons as opposed to 0.95? Unless recordings are performed from downstream neurons that cannot really be determined. Both 0.8 and 0.95 face the problem of ambiguity of whether it is interpretable by a downstream neuron, however, a probability of 0.95 will enable readers to judge what happens when events are statistically significant. Can the authors discuss this in the main text?

9. Very low decoding errors (3-6 cm) during the run at the small timescale employed here (2 ms) may indicate that this method is incapable of capturing theta sequences (which should give a higher decoding error). Instead, they capture stationary dynamics like they do during rest. Larger bins for decoding during run are typically used to study position representation at larger timescales (at bigger timescales than theta sequences) which are likely to have low error. This could be problematic since the current method might be biased toward over-representing the stationary dynamic. The authors should address this in the manuscript.

10. What happens when a 20 ms time bin is used for state space modelling? Does it reduce the proportion of spatially-coherent events?

11. Figure 5. When whole ripples are classified as stationary, stationary-continuous mixture and so on, how is this classification done for the whole ripple? The classification of dynamics is done for each 2 ms bin, but what criteria are used to classify the whole ripple? Please provide more details on the intermediate steps. For instance, if an event is stationary, but only has one 2 ms bin which deviates, is it considered stationary-continuous?

---

## [Author Response]

Essential revisions:1. The authors should further test their method on exploration data to validate the approach, especially since non place cells may potentially bias the decoding of animal's speed in clusterless data (see reviewer #3 point #1). This analysis of hippocampal activity during run can be used to identify the encoded speed in within theta sequences (reviewer #1, point #2).

We have added a comparison between the decoded position and the animal’s position to validate our decoding of speed (page 7) as well as added a discussion about the ability of our model to identify speed during theta sequences. We have provided a more detailed response to reviewers #3 and #1 below.

2. The authors have to improve controls, especially their shuffling procedures. Shuffling the spike-location relationship is too permissive. See reviewer #2, point #1 and reviewer #3, point #9.

We have added an additional shuffle where we randomized the order of runs (from one reward well to another) and then circularly permuted the resulting segments of data across all tetrodes uniformly (page 9). This shuffle preserves local spiking correlations (and thus spiking statistics), but the relationship to position is disrupted. We find that our results are very different when our model is applied to these shuffled data as we describe in detail in our response to reviewers #2 and #3 below.

3. The authors should use more stringent thresholds on speed and theta power to rule out the possibility that some of the events are not replays but represent current position. See reviewer #2, points #2-3.

We have performed a control analysis where we required animal speeds to be less than 1 cm/s and achieved similar results (Figure5—figure supplement 2C). We also show in Figure 5 that most of the dynamics do not represent the animal’s current position (although the stationary dynamics often represent the animal’s position). We have provided a more detailed response to reviewer #2 below.

4. The classification of candidate replay events should be improved (especially events that may be partially "unclassified") and the authors should provide a more detailed comparison of the replay events detected with their method and with previously described methods (reviewer #3, points #2-3)

We included more information about which SWR included times that were unclassified (Figure 5). We also performed a comparison between our method and three previously described methods (Figure 6). We have provided a more detailed response to reviewer #2 below.

5. The authors should provide descriptive statistics demonstrating that spiking properties during candidate replay events (e.g. average rate per tetrode) are similar between sorted and clusterless data (reviewer #3 point #6-7). Only events in which spikes are detected on multiple tetrodes should be included, to prevent decoding from the firing of a single cell. Finally, spikes of putative inhibitory neurons should be discarded (reviewer #3, point #7).

We have included descriptive statistics about the spiking statistics during each of the dynamics (Figure 5—figure supplement 3) as well as performed a control analysis using clustered units (Figure 5—figure supplement 2D). We have clarified in the Methods (page 22) that all the SWR events that we have analyzed included more than 2 tetrodes with spikes and we have included a further control where we required more than 5 tetrodes with spikes (Figure 5—figure supplement 2A). We have also performed a control analysis where we removed spikes with narrow waveforms to exclude spikes from putative interneurons (Figure 5—figure supplement 2B) and another analysis where we included only sorted putative pyramidal cells (Figure 5—figure supplement 2D). We have provided a more detailed response to reviewer #3 below.

6. The authors have used neuronal data collected from the different subcircuits of the hippocampus (CA1-3). It should be demonstrated that the main results hold if the analysis is restricted to CA1, on which most of previously published studies are based, especially since it can affect the nature of replay events. Is there a relationship between the category (e.g. continuous or stationary) of a given replay event and the proportion of neurons from the different substructures that fired during that event? Also, discussing the seemingly low proportion of continuous events (if the same proportion is observed when only including CA1 neurons). See reviewer #2, point #4 and reviewer #3, points #4-5.

We have performed a control analysis where we include only CA1 tetrodes (Figure5—figure supplement 2E). Our dataset does not allow comparison of hippocampal subregions because of the varying number of tetrodes in each region. We have also included a discussion (page 18) of why our proportion of continuous events seem lower than those observed in the literature, but are actually in line with the proportion observed in the literature because of the capability of our model to characterize more than one dynamic per SWR. We have provided a more detailed response to reviewers #2 and #3 below.

7. The title of the manuscript is a bit misleading as the reported replayed speeds are often even slower than "real-world speed" and should therefore be changed. See reviewer #1, point #1 and reviewer #3, point #8.

Here we respectfully disagree. We now provide a plot showing the actual speeds at which the animal moves through the environment, which cover a similar range to those seen during the slower replay events (Figure 5G). We have provided a more detailed response to reviewers #1 and #3 below.

8. The authors should further discuss the implication of their findings. See reviewer #1, point #3 and reviewer #2, point #4.

We have further expanded our discussion to discuss the functional implications of the slower replays and the possible subcircuits that could underlie the different replay speeds within the hippocampal structure (page 19). We have provided a more detailed response to reviewers #1 and #2 below.

Reviewer #2:Denovellis and colleagues develop and apply a state space model to decode position representation during hippocampal SWR events. The main advantage of this model is that is makes fewer assumptions of the dynamics of SWR events allowing it capturing varying and mixed movement dynamics. Through the application of this model the authors are able to decode coherent spatial content from the vast majority (~90%) of SWR events – a much, much higher proportion than previous papers on the topic have reported – and they show that, contrary to previous work, the most common type of movement dynamic during SWR events is one that's characterised by a stationary/slow movement speeds.I believe the work by Denovellis and colleagues represents a significant advance in our understanding of SWR events. Moreover, the state space method Denove et al. described also represents useful analytical tool which could potentially be useful to many neuroscientists in the field. I would also like to commend the authors for their clear and detailed explanation of the model.I do however have some concerns about the robustness of their SWR event method and some of their control analyses. Below I explain these in more details and suggest ways to address these concerns. All suggestions are implementable and would, I believe, significantly strengthen the authors' conclusions.1. To control for spurious SWR events (containing spatially coherent content), the authors shuffle the position-spike relationship. Although this type of control analysis is commonly used, it can be overly permissive, particularly if researchers are working with small-to-medium samples sizes. This is because for smaller samples sizes it is unlikely that all positions are represented equally by the cell population (which is an assumption of the control analysis). However, given the authors are using clusterless decoding, a cell ID shuffle is not possible. Thus to address this potential caveat, it would be useful if the authors show position representation in their cell population on the W-maze? Moreover, if the position representation is not equal throughout the environment the authors need to account for this in their position resampling with replacement control analysis.

We thank the reviewer for this comment. As the reviewer noted, it is important to know how well the spatial environment is represented by cells because this could affect the decode of SWRs. To address this, we have added Figure 2—figure supplement 3 to show that the firing over all our tetrodes is roughly equal for most of our epochs. We believe this is one of the advantages of clusterless decoding in that we have more spikes and therefore are potentially less constrained by worries about under sampling positions in the environment. Furthermore, we also measured the median difference between the most probable decoded position and the animal’s actual position and found a 4 cm median difference (3-6 cm 95% CI), for all time in sessions. This gives us confidence that the cell population has spatial information about all positions in the environment.

We would also like to note that we directly test for non-spatially coherent content by including the fragmented dynamic in our model, which is what the shuffle tests are aimed at capturing in most analyses. We have added to the Results to clarify this point (page 6).

In addition, to better illustrate this point, we have added an additional shuffle where we have randomized the order of the runs between wells and circularly permuted the spatial position within the runs to preserve more of the structure of the task (Figure 3—figure supplement 3, page 9). This shuffle increased the prevalence of spatially incoherent dynamics and decreased the prevalence of spatially coherent dynamics as expected, showing that our addition of the fragmented state works well to capture these types of dynamics. The shuffle also substantially increased the spatial extent of the 95% HPD region, indicating that the model was much less certain of the decoded locations even in cases where it identified a dynamic. These results indicate that our results reflect structure in the data, not just structure in the model.

2. The authors use a speed threshold of 4cm/s to exclude SWR events that may occur during movement periods. Could the authors use a more stringent threshold – e.g. 1cm/sec or 0cm/sec? Although 4cm/sec is low I believe it is important to rule out the possibility that some of their detected spatially coherent SWR events don't merely reflect the movement of the animal during the event. This is particularly important given their finding that the most common type of replay event is one that is characterised by slow movement speeds – akin to those seen during actual behaviour.

As per the reviewer’s suggestion, we reran our analysis using a speed threshold of less than 1 cm/s and found a similar distribution of movement dynamics (Figure 5—figure supplement 2C). Also note that the majority of our SWRs occur at speeds less than 1 cm/s (Figure 5—figure supplement 3B). We have also quantified the distance of the decoded trajectory from the animal for stationary-continuous-mixtures (Figure 5C) and found that these represented locations that were some distance away from the animal (median 52 cm). Because of this we are confident that our results are not simply encoding the animal’s position or the result of theta sequences as the animal begins to move.

3. Related to this, the authors could also use low theta power as an additional quality criteria to weed out events that may reflect (slow) movement periods.

We agree that this is a very reasonable suggestion, and in fact we have tried this in the past. Unfortunately, low theta power is difficult to use as a criteria because the sharp wave that occurs during the SWR is also in this same frequency band. In our experience, there is no consistent threshold of theta power (or even theta/δ ratio) that consistently separates immobility from movement times. Because of these challenges, and because of our analysis above, we feel confident that our results are not due to movement of the animal and slow theta sequences.

4. One of the main findings of the paper is that SWR events can consist of different types of movement dynamics (stationary, continuous, fragmented) or a mixture of dynamics. This is indeed an interesting and novel finding, but it would be useful if the authors could elaborate on the theoretical implications of different type of replay events. For example, do the authors think SWR events belonging to different movement dynamic categories are supported by different sub-circuits within the hippocampus? For example, some work has shown SWR events preceded by EC activity are longer in duration Oliva et al. (2016).

As per the reviewer’s suggestion, we have broadened our Discussion to talk about the theoretical implications of the different types of replay events (page 19). In particular, for the slower dynamics: stationary dynamics could be important for recalling specific locations in the environment and stationary-continuous-dynamics could be useful for recalling or evaluating small snippets of the environment. Continuous dynamics could be useful when needing to recall or consolidate an entire trajectory. Finally, fragmented dynamics could correspond to non-spatial memories or memories of other spatial environments. When there are jumps between positions, perhaps these represent instances when several snippets are consolidated in succession.

For example, do the authors think SWR events belonging to different movement dynamic categories are supported by different sub-circuits within the hippocampus? For example, some work has shown SWR events preceded by EC activity are longer in duration Oliva et al. (2016).

It is certainly possible that the different dynamics are driven by different subcircuits within the hippocampus because awake SWRs are influenced by input from outside of the hippocampus, as the reviewer noted. However, due to the varying number of tetrodes in each hippocampal subfield, we are unable to attribute particular dynamics to specific subregions. We have added this to the Discussion (page 19).

Reviewer #3:The authors of this study investigated the prevalence and range of representational dynamics associated with replay activity in the rat hippocampus during periods of rest following a spatial alternation task in a W-shape maze. Using a state space model that captures the spatial content and temporal evolution of replay during sharp-wave ripple activity, the authors report that most ripple events contain spatially interpretable content that often progresses at timescales slower than previously reported and that appears to better match previous wake activity at real time. The reported findings challenge the classical view of replay as primarily time-compressed representation of previous wake activity and build on previous work on stationary events and reduced proportion of significant trajectory replay events to provide a fuller picture of the repertoire of possible content for ripple events during rest. While the results are intriguing, a number of conceptual and technical shortcomings prevent a more enthusiastic appreciation of this work in its current state, as detailed below.1. The main Bayesian model needs to be tested first during run on track, where the prior is being built. The authors need to compute the error of the model during run in terms of the difference between the decoded position during run and the actual animal trajectory. If, for any reason, the model appears stationary during run at any trial/lap, then the authors will need to take that into account when claiming stationarity during rest. In particular, clusterless decoding may make use of spikes that are not spatially well modulated and that would appear to support the stationary events during rest.

As per the reviewers suggestion, we computed the median absolute difference between the animal’s most probable decoded position during run and the animal’s actual location, using a five fold cross validation. We found that the median difference between these two quantities was comparable to other studies (4 cm median difference, 3-6 cm 95% CI, for all time in sessions), even though we used 2 ms time bins, rather than the more commonly used 250 ms time bins. We have added this to the Results (page 7). We also note that the slower spatially coherent trajectories had posteriors that were highly spatially concentrated (as characterized by the size of the 95% highest posterior density in Figure 4E). This can only happen if the spikes themselves were from cells that were strongly spatially modulated, ruling out the possibility that the stationary events are driven by cells lacking spatial selectivity.

2. Please use more stringent classification of the events.

As suggested by the reviewer, we have added more comparison between our method and previous methods in the Results (Figure 6). However, we believe that it is a strength of our decoder that the classification of dynamics is not dependent on the entire SWR. The determination of the start and end times of the SWR are somewhat ad hoc and vary greatly from paper to paper. Because of this, other studies have used a procedure where the stop and start is determined by the decode and not the multiunit or SWR event boundary (Xu and Csicsvari 2019, Pfeiffer and Foster 2015). Using the whole SWR for determination of the trajectory means any noise or spatially incoherent period during the SWR could cause the whole event to be thrown out and not analyzed. Our method allows us to characterize the content of all SWRs, rather than exclude certain SWRs because the SWR does not fit our assumptions about constant velocity trajectories. Furthermore, we have shown that the spatially coherent dynamics that we characterize are sustained in duration (Figure 5, stationary-continuous-mixtures have a median duration of 73 ms for example). Finally, we and others have shown examples where there are multiple dynamics in a single SWR (Figure 3, Figure 3—figure supplement 1, Pfeiffer and Foster 2015), so we think it would be inappropriate to classify these types of events as a single dynamic. We have clarified this in the manuscript (pages 11, 18).

As the current relaxed criterion for classifying events only requires a portion of the event to exhibit that behavior, what proportion of events classified as 'stationary' or 'continuous' significantly pass traditional, rigorous event-matched shuffles?

As requested by the reviewer, we have performed the comparison of our dynamics to the standard Bayesian decoder that passed event-matched shuffles (page 15). These were all based on a Radon line fit with 20 ms bins, have a replay score with p < 0.05 using a circular shuffle of the posterior values for each time bin, and use clusterless decoding. We chose this method as the “traditional” method, but please note, as we mentioned in our Introduction, that there are many other methods that have been used in the replay field. We would like to emphasize that the dynamics captured by our decoder are not required to be present for the entire SWR while the line is fit over the entire SWR, so the comparison is, in our opinion, hard to interpret. However, we agree with the reviewer that, despite these limitations, it is useful to have this comparison for others in the field to understand the differences between the methods, so we have included them here and in the manuscript (Figure 6):

– 8% (542 of 7176) of SWRs containing stationary dynamics with p < 0.05.

– 14% (2253 of 16106) of SWRs containing stationary-continuous-mixtures dynamics with p < 0.05.

– 31% (1368 of 4511) of SWRs containing continuous dynamics with p < 0.05.

– 31% (799 of 2562) of SWRs containing fragmented-continuous dynamics with p < 0.05.

– 15% (244 of 1598) of SWRs containing fragmented dynamics with p < 0.05.

What proportion of extended stationary events, >100 ms for instance, have a line-fit replay score above the 95th percentile of event-matched shuffles (circular shifting of posterior decoded probabilities) and a trajectory length shorter than a certain value, e.g. 6 cm?

28% of extended stationary events have a line-fit replay score with p < 0.05.

Similarly, what proportion of continuous trajectories pass their respective event-matched shuffles?

30% of SWRs with continuous dynamics have a line-fit replay score with p < 0.05.

What proportion of stationary events are entirely stationary? The authors need to quantify all of the above and present them in the Results.

47% of SWRs containing the stationary dynamics are entirely stationary. We have added all of these quantifications to the results.

3. One significant conclusion of this paper is that most population events during awake rest are stationary-continuous mixtures or stationary as opposed to continuous (replay) events. However, given that an event is classified as stationary if part of the event is stationary (probability >0.8), there is a possibility that the rest of the event is unclassified (no clear structure). In plots 3G, 5A-F it is important to include details about the unclassified ripples and the unclassified parts of the ripple which is otherwise classified. For instance, in 5A, the proportion of ripples with parts that are unclassified needs to be mentioned. This is important because parts of the event might be stationary, and parts might be noise/unclassified.

We agree with the reviewer that including information about the unclassified times is important in order to fully characterize the SWRs. As suggested, we included the unclassified categorization in Figure 5 to give the readers a sense for how often the unclassified category occurs, the duration of these time periods, and the multiunit firing during these time periods. We have also included the unclassified category in Figure 5—figure supplement 3 to add information about the ripple power, spikes per bin, and number of spikes per tetrode.

Previous studies have characterized events as stationary only if they were stationary throughout the event and had excess stationarity than the 95th percentile of an event-based shuffle. How many of the events classified as stationary pass this significance criterion?

5% of events classified as stationary were significant at the 0.05 level using the standard Bayesian decoder with Radon transform line fit. We feel that our findings argue strongly against trusting this 5% number as an indication that there are no meaningful stationary events.

This relaxed classification might otherwise lead to over-classification of certain event types.

We hope that we have addressed this issue in the answers above and in the text (page 15), and we want to emphasize that given that many events contain multiple dynamics, the standard line-fitting approaches are problematic in terms of accurately capturing the way that representations move during these events. The goal of our approach is to provide information about the sorts of representational dynamics that are present and that could provide information to downstream structures, and for that goal, we felt that a threshold of 0.80 was reasonable. That said, future work that examines how these events influence activity and plasticity should help us refine these classifications further.

4. The reported results are based on combined recordings from 3 hippocampal areas: CA1, CA2, and CA3 (conform Methods). These areas have different cell types, some of which respond to animal stationarity rather than movement. This makes classification and comparison of all event types between this study and the previous studies reporting on CA1 place cells only which were active on simpler tasks on linear tracks difficult. The authors should restrict their analysis to tetrodes located exclusively in CA1 when computing proportions of event types and only then compare their results to previous studies reporting on significant replay and stationary events.

As requested by the reviewer, we performed a control analysis where we only used CA1 tetrodes for clusterless decoding. We found similar dynamics as reported in the main result of the paper (Figure 5—figure supplement 2E).

However, we would like to point out that several studies (e.g. Diba and Buszaki 2007, Karlsson and Frank 2009) investigating replay have used cells from multiple subregions within the hippocampus. Second, we have also found cells show place specific activity during immobility across CA1, CA2 and CA3 (Yu et al. 2017, Kay et al. 2016). Finally, we have also found that CA1 and CA3 spiking are largely coordinated within and across hemispheres during awake SWR replay (Carr et al. 2012). For these reasons, we feel our main analysis should not exclude CA2 and CA3 tetrodes.

To make this comparison even more interpretable, similar significance tests against shuffles and use of clustered data (available for 9 out of the 10 animals that were previously published) should be performed. If the presented proportions of events persist under these conditions, the impact of the findings will be significantly increased.

We appreciate this point, and carried out an analysis of replays events based on clustered data. The results were very similar (Figure 5—figure supplement 2D).

5. There is discrepancy in proportion of replay events reported here compared to previous studies. Continuous events are most similar to traditionally detected replay sequences, however, those sequences are unidirectional (move from one side of the track to the other to be significant), while these events can move in both directions (model of movement dynamic is symmetric for continuous). The authors should investigate and explain why the proportions of continuous events are less than those of traditionally detected replay sequence events with event-matched shuffles (10-50% in other studies as stated by the authors versus ~5% only continuous).

The reviewer brings up an important point that we failed to clarify in the text. We believe that there is not a discrepancy in the proportion of replay events in our study compared to previous studies. Our decoder is much more flexible in terms of allowing different speeds during the SWR event, so an event that might have been characterized as having constant high speed for the entire event might actually have some time periods that have slower trajectories, even if the majority the time the trajectory proceeds at a higher speed (see Figure 2 for example). Accordingly, we still find 19% of classified SWR events contain continuous trajectories, which is consistent with those previous studies, and these continuous trajectories persist for a considerable duration (median 94 ms, Figure 5B). As the reviewer noted, only 5% of SWRs have continuous dynamics throughout the entire duration of the SWR, but we feel that restricting our analyses to dynamics which last the entire SWR is a limited characterization of the SWR, particularly given that we observed SWRs with multiple dynamics. Finally, we have also used a less strict threshold for including SWRs and interpreted a much larger fraction of these SWRs, so the percentage may be hard to compare between studies. We have added this clarification to the text (page 18).

6. The properties of each event type need to be compared to rule out other reasons for their distinct spatial and temporal dynamics. These properties to be compared between classes of event types (e.g. stationary, continuous, etc.) include; number of tetrodes active in the event, ripple power, total number of spikes in the event, spikes per bin, event duration, proportion of spikes under 6 ms ISI for each tetrode in each event type, proportion of the event duration it lies in that classification.

We thank the reviewer for this suggestion. We have added these statistics as a supplementary figure (Figure 5—figure supplement 3). These statistics confirm that slower dynamics are not driven by a single neuron as there are many tetrodes active for each dynamic and there are multiple spikes from different tetrodes in each time bin. There is also little evidence for more bursting in the slower dynamics compared to the continuous dynamics because we observed a similar proportion of spikes under 6 ms ISI for the slow dynamics. Only ~50% of the spikes had ISIs of less than 6 ms in the stationary, stationary-continuous, and continuous dynamics (Figure 5—figure supplement 3). It should be noted that this metric should be interpreted with extreme caution because with clusterless decoding, the ISI of the multiunit spikes is not the same as if we had an individual unit. The spikes could be from multiple neurons firing, which would not tell us if there was bursting or not. Finally, these statistics show similar ripple power for all our dynamics as well as a tendency for the unclassified time periods during the SWR to have lower average ripple power.

Could you also confirm that population events included for clusterless decoding are similar in properties to those obtained from clustered data by comparing their properties (duration and z-scored population firing rates across the event)?

We apologize for the confusion regarding our event detection method. In this manuscript, we analyzed SWR events which are detected based on the combined instantaneous ripple-band power of CA1 LFPs and not based on high multiunit firing rate, as many studies have done (Xu et al. 2019, Farooq and Dragoi 2019, Drieu et al. 2018, Ólafsdóttir et al. 2017, Grossmark and Buzsáki 2016, Pfeiffer and Foster 2013, Davidson et al. 2009). Therefore, we are not able to compare the multiunit and clustered events because they are not selected on this basis.

7. There is a possibility that one neuron dominates a population event and, therefore, stationary events are simply single units firing continuously. For instance, Figure 1A for the simulation shows just that, one simulated neuron fires and a stationary event is predicted by the algorithm. As clusterless decoding is used for the main analyses, can only events consisting of multiunit activity from at least 5 tetrodes be included in the analysis? Will the main classes of events be replicated under these conditions?

As suggested by the reviewer, to exclude the possibility that a single neuron firing is the cause of stationary events, we performed a control analysis where we required spiking from at least 5 tetrodes for a SWR to be analyzed and found a similar distribution of movement dynamics (Figure 5—figure supplement 2A). It also should be noted that our original analysis required at least 2 tetrodes to be active for a SWR to be analyzed and we have clarified this in the Methods (page 22). We have also included a histogram of the number of active tetrodes per SWR event (Figure 5—figure supplement 3B) as well as a boxplot of the number of active tetrodes by dynamic (Figure 5—figure supplement 3B). These show that well over 5 tetrodes are active in each dynamic and during the SWR events.

Related to this, interneurons (putative inhibitory neurons) should be removed from the clusterless decoding based on waveform shape as there is a possibility that inclusion of interneurons in the clustering decoding analysis biases results towards detection of stationary events (there might be a hint of this from examples presented in Figure S4 where good stationary events seem to have higher spiking per tetrode compared to continuous events). The event classes should be recomputed and compared with the original classes.

As requested by the reviewer, we performed a control analysis where we excluded spikes with narrow waveform spike widths (<0.3 ms), which would preferentially exclude spikes from putative interneurons. We again found a similar distribution as in our main results (Figure 5—figure supplement 2B). However, we should note several caveats to this approach. First, interneurons are typically identified by both their high firing rate and the average narrow spike width waveforms. Our approach only considered the spike width for each individual spike (not average) and therefore may be much less discriminating and remove many spikes that may contain spatial information. Second, it is not clear that interneurons should be excluded. Interneurons can have spatially specific firing and may contain information about the position represented by the hippocampus (Hangya et al. 2010, Wilent and Nitz 2007). Third, if interneurons were not spatially specific and driving the slower dynamics, we would expect to see the posterior be less spatially concentrated. Instead, we found that these dynamics tended to have narrow posteriors (as characterized by spatial coverage of the 95% highest posterior density values in Figure 4E).

Finally, we performed a third control analysis where we also looked at the 9 of 10 animals in our dataset that already had clustered cells, excluding putative interneurons based on spike width and firing rate and requiring that at least three putative pyramidal cells be active during the SWR. Although there were many fewer events to examine, we still observed many stationary and stationary-continuous-mixtures in SWRs compared to continuous sequences (Figure 5—figure supplement 2D).

8. The current title is misleading. There does not appear to be clear evidence of replay at real-world speeds. First, during each run lap, the animal trajectory changes unidirectionally at an estimated speed of 25-35 cm/s. For replay to occur at real time speed, the trajectory decoded during rest should fully unfold at the same speed over 3-4 s for a 1m-long track. This is never reported here, partly because the events are shorter. The stationary events decode individual locations, but there is no indication that they evolve in succession (successive locations at successive times) to represent trajectories at real time speeds.

We thank the reviewer for this comment. As the reviewer stated, the slower trajectories cannot span the same amount of distance as when the animal is moving at slow speeds because of the duration of the SWR. Our claim in the title and in the rest of the manuscript is that the speed of the decoded trajectory over the duration of the SWR is often similar to the speeds at which the animal experiences the environment, which includes times when the animal is immobile and times when the animal is running. This does indeed imply that the replay trajectory does not move very far. We have clarified this in the text (pages 12) and added a figure showing the distribution of speeds of the animal (Figure 5G). As this figure shows, there is substantial overlap between the movement speeds seen during replay and real movement speeds.

Second, neurons fire in time-compressed theta sequences during run, so neuronal activity is naturally compressed even before replay. Compressed replay essentially already represents space at roughly the same speed as during run (theta sequences).

The reviewer brings up an interesting and important point. Individual theta cycles can contain extended sequences spanning locations behind and in front of the animal, but other theta cycles can show much less prominent sequences, resulting in a range of potential representational movement speeds. In addition, a recent result reported quite different dynamics for the first and second halves of theta cycles (Wang et al. 2020).

Whether replay sequences are essentially the same as theta sequences remains to be determined, but we have noted that our methods could be used to study these sequences as well.

We emphasize, though, that the goal of this paper is to establish that replay sequences can proceed at a variety of speeds. We hope in future manuscripts to directly address the similarities and differences between replay and theta sequences using our approach.

9. Bursting of neurons or other individual neuronal properties might lead to classification of successive bins as spatially coherent and/or stationary. In order to control for that possibility, in clustered decoding researchers typically generate shuffled datasets with conserved place map properties by circularly shifting individual place maps. The shuffles generated in this paper randomize the position to spike relationship destroying all single-cell properties. Can shuffles with conserved tetrode multiunit activity properties be generated instead to test this hypothesis? E.g. circularly shift by the same value all spikes from the same tetrode.

The reviewer brings up an important point, which highlights our use of small time bins. Our time bins are 2 ms, which implies spike bursts (3-8 ms ISI, 2-3 spikes per burst, Tropp Sneider et al. (2006)) could fall in separate, potentially successive time bins as the reviewer noted (although the average ISI is ~6 ms so it’s likely not the case, Tropp Sneider et al. (2006)). Note that our use of 2 ms time bins is an advantage over the standard 20 ms time bins because with 20 ms time bins, either most of these spikes fall into one time bin, making the estimate stationary over the 20 ms time bin or the spikes could fall into two 20 ms time bins, making the estimate stationary over 40 ms; with our approach, the stationary period could only last for approximately as long as the burst modulo 1 or 2 ms.

As the reviewer stated, spike bursts from a single neuron could lead to successive time bins being classified as stationary. There are two lines of evidence that make us think that this is not the case. First, the stationary dynamics have a longer duration than one would expect from a burst alone (54 ms median duration, 38-74 ms quartiles vs. bursts which have a duration of 6-24 ms). Second, thanks to a previous suggestion of this reviewer, we know that multiple tetrodes are active during each SWR (Figure 5—figure supplement 3B), there are often multiple spikes from different tetrodes in a given time bin (Figure 5—figure supplement 3A), and the proportion of spikes under 6 ms for stationary dynamics is similar to those as stationary-continuous-mixtures and continuous dynamics (Figure 5—figure supplement 3A). Moreover, our main claim in the manuscript is that the stationary-continuous-mixtures constitute the bulk of dynamics in SWRs, which can only happen with more than one tetrode active.

[Editors' note: further revisions were suggested prior to acceptance, as described below.]

The manuscript has been improved but there are some remaining issues that need to be addressed, as outlined below:The most critical aspect is about the decoding. During wake, theta sequences should strongly impact the accuracy of the decoding with 2m time bins, but it is unclear why this is not the case.

The number we reported included both times when the animal was immobile and times when the animal was running. Because of this, the decoding accuracy was not strictly computed on times when theta was most prevalent. To prevent confusion for the readers, we have changed the decoding accuracy to be measured only on the times when the animal was moving. This resulted in a change of decoding accuracy from a median error of 4 cm *(3-6 cm) to* 7 cm (5-9 cm), which is in line with most previous results. We also note that we show an example of decoding during movement in a separate manuscript that uses a version of this algorithm (Gillespie et al. Biorxiv 2021, Figure 2B) which illustrates its capacity to detect theta sequences.

The study should also provide some comparison of decoding accuracy with 2ms and 20ms and provide further details regarding how exactly events are eventually classified (see reviewer #2, points 9-11).

We clarified this issue regarding decoding accuracy with Dr. Peyrache and have therefore not included this first analysis.

For the second analysis we now provide further details on the classification of events on page 26 and in our response to the reviewer. We also clarified this issue with Dr. Peyrache.

The study should also include examples showing the differences and similarities in decoding with the clusterless and more classical approaches (see reviewer #2, point #1).

We have added six examples comparing the differences in decoding with clusterless and more classical approaches in a new supplemental figure: Figure 6—figure supplement 1.

Some other concerns were raised, mostly requesting clarification in the methods and the discussion of the results. Please see detailed review below (reviewer #3, points #2-8).Reviewer #3:In the revised manuscript, the authors have partially addressed the previous concerns. The manuscript has improved, and the findings remain potentially interesting. Below, please find additional comments aimed at improving the accuracy and readability of the manuscript:1. Due to the various differences from earlier methods (clusterless decoding, state space encoding model, bin size, event detection criteria and quantification of dynamic type: radon transform vs current method), it is difficult to pinpoint which of these differences is leading to the authors' main conclusions. For the general readership, it will be extremely useful to show actual examples of the same decoded spike trains using standard Bayesian decoding (with clustered units) and using state space models. This could reveal any significant differences in the decoded locations which the authors claim are revealed by this method (pages 3 and 4). Since the authors claim there is a substantial difference between the two methods, differences in the inferred dynamics from the two methods should be visible by eye and many such examples should be available.

We thank the reviewer for this great suggestion. We have added six examples of the fits from three different methods in a new supplemental figure: Figure 6-supplemental1. These examples illustrate the variety of event types seen and the challenges inherent in using 20 ms bins and standard “Bayesian” decoding approaches.

2. Is the higher preponderance of stationary/slow events caused by the authors setting of the beginning and ends of events as the start and ends of ripples, rather than using the multiunit population activity (which can span multiple ripples). Can the authors rerun the analysis with multiunit population activity to ensure that the results are not different from previous studies mostly due to how events are detected? Even if the results are due to that, it will be useful to report that to the reader as it will not diminish the impact of these findings (the authors already allude to the fact that the cut off is arbitrary) but will help them understand the reason for the differences between the studies.

As requested by the reviewer, we reran the analysis using times of high multiunit activity during immobility and found that the distribution of slow dynamics is qualitatively similar. Thus, the preponderance of slow events does not seem to be a result of using sharp wave ripple rather than events with high multiunit activity. We have included this as Figure5—figure supplement 2F in the manuscript. This also corroborates our finding that these slow dynamics have similar multiunit firing rates to the higher speed, continuous dynamics (Figure 5F).

3. Can the authors include a histogram of event duration versus proportion of such events for each of the dynamic so that the reader can better judge what is the likelihood of each event type by duration?

We are unclear on whether proportion in this case refers to proportion relative to the length of the sharp wave ripple or whether proportion refers to over a recording session. We have included both here as Author response image 1 and Author response image 2. Given that we already show the distributions of dynamics per SWR and the distributions of the durations of the dynamics in main figures, it is not clear to us that these Author response images will substantially augment a reader’s understanding of our results, so we have not included them in the manuscript, but if the reviewer feel strongly that they need to be included we can do so.

**Author response image 1. respfig1:** Proportion of SWRs for each day vs duration of the dynamic within the SWR. Each dot represents one day for one animal.

**Author response image 2. respfig2:** For each dynamic, the duration of the dynamic vs the proportion of time of the total SWR duration.

4. Although the fragmented dynamic would indeed identify discontinuities in decoded locations, how exactly is it better than a non-parametric shuffle? What if the fragmented dynamic has more false-positives or false-negatives?

All statistical methods have trade-offs. However, we believe that our fragmented state directly tests for “random” on a moment-by-moment basis. This is an advantage because non-parametric methods cannot do this on a moment-by-moment basis thereby potentially excluding content of interest. Shuffles also can require much time for computation often leading to small numbers of shuffles, which can make the tests underpowered. Finally, shuffles can lead to pathological null distributions which cannot occur in real data and therefore do not test against the null hypothesis of interest (e.g Foster 2017, van der Meer et al. 2020). Thus, including the fragmented dynamic preserves the existing data while still testing for the different dynamics. More broadly, testing for false-positives and false-negatives requires a model of what a true positive and a true negative is. The state space approach provides a relatively simple, interpretable version of such a model, while the model associated with a shuffle is often harder to define and understand. We have added this point to the Discussion.

5. Are the discovered dynamics restricted to the replay of the animal experience or can they be present before the experience? The authors should discuss this possibility.

Our study was not focused on these types of events but indeed our model could be used to discover dynamics present for the experience. We added this point to the Discussion that we could use this model to detect dynamics before the event (page 20).

6. How is stationarity a real-world 'speed'? Isn't it a lack of movement? Were only the spikes emitted while the animal was moving used for the encoding model and not those while it was resting? How are then the stationary events replays of that movement? What was the average velocity of the animal during movement? And of the stationary dynamic?

To answer the first question, the animal’s experience is not limited to running. The animals can stop at any position on the track, and thus a real world experience can include movement and immobility at any given position.

We only used spikes that occurred in association with speeds of > 4 cm / second in the encoding model. We had mentioned that in the Methods for the single spike analysis, but we had neglected to do so for the clusterless analysis, so we have added that information. We note, however, that place specific spiking in a location can continue even when animals are still (Kay et al. 2016; Yu et al. 2017), so using this encoding model still allows for the identification of periods of immobility or slow movement in a decoded representation. More importantly, stationary events are those that have slow velocity, but nowhere do we claim that this velocity is < 4 cm/second. Indeed, given our bin size of 3 cm, a 100 ms event could be moving at (for example) 10 cm /second and still remain within a single bin. Thus, we only claim that these events represent very slow speed movements consistent with (slower) real-world speeds, not that they are perfectly stationary at a point in space.

Finally, the median speed of our animals is 4 cm/s for all times (including immobility times) and 17 cm/s for run periods (speeds > 4 cm/s). We have added this to the manuscript in the Results (page 14).

7. References cited and conclusions drawn from papers are not accurate in some instances, please verify and correct. Several cases are listed below, but the list is larger. Louie and Wilson, Neuron 2001 showed that replay of experience can occur at real-time speed during REM sleep, which should be cited.

We thank the reviewer for the reference. We have added the reference to Louie and Wilson 2001 on page 18 in the Discussion.

On lines 67-68 the authors say: "events that represent a single location are only seen in young animals (Stella et al., 2019)". That study did not investigate young animals, the authors probably meant to refer to (Farooq and Dragoi, 2019), which should be cited.

We thank the reviewer for catching this mistake. We have changed the reference to Farooq and Dragoi (2019) as we originally intended, which was erroneously referenced as Stella et al. 2019. We have correctly referenced Farooq and Dragoi (2019) in other parts of the paper and apologize for our carelessness in this instance.

There is a difference between what previous studies have claimed, and what the authors attribute to those studies. For instance, previous studies observed low proportions of sequential trajectories, but they did not observe low proportions of spatially coherent time-bins/cofiring, which are otherwise well-reported and have been observed many times.

We agree that previous studies that examined co-firing across entire events found substantial structure, but those studies did not determine whether individual events were expressing spatially coherent structure across a set of time bins. We have modified the introduction and discussion to reflect this clarification.

8. How do the authors know that a probability of 0.8 or above can be inferred from downstream neurons as opposed to 0.95? Unless recordings are performed from downstream neurons that cannot really be determined. Both 0.8 and 0.95 face the problem of ambiguity of whether it is interpretable by a downstream neuron, however, a probability of 0.95 will enable readers to judge what happens when events are statistically significant. Can the authors discuss this in the main text?

Our threshold of 0.8 is used to categorize the dynamics to determine the speed and as a measure of our confidence in that dynamic. We are not claiming downstream neurons are explicitly making use of our dynamics categories, but rather our categories can be used to understand and summarize the dynamics of replay. Thus, our algorithm enables the possibility of a systematic investigation of how different thresholds and thus different category boundaries might be related to activity in downstream regions.

We have also shown that these categories broadly correspond to speeds of interest in Figure 1F. Additionally, we have also shown that our results still hold with a higher 0.95 threshold. We have added a discussion of this on page 6.

9. Very low decoding errors (3-6 cm) during the run at the small timescale employed here (2 ms) may indicate that this method is incapable of capturing theta sequences (which should give a higher decoding error). Instead, they capture stationary dynamics like they do during rest. Larger bins for decoding during run are typically used to study position representation at larger timescales (at bigger timescales than theta sequences) which are likely to have low error. This could be problematic since the current method might be biased toward over-representing the stationary dynamic. The authors should address this in the manuscript.

We reported decoding error for both immobility and running time periods to show that our model could capture stationary representation of the animal’s position at the well as well as theta sequences when the animal was running. We realize, however, this is confusing to the reader given what is typically reported. We have changed this to be the error only during running times. During running, we observed a median error of 7 cm (5-9 cm), showing that we indeed can capture theta sequences. This is similar to most other studies (Davidson et al. 2009: 7 cm, 9 cm, 8 cm, 8 cm for each rat; Shin et al. 2019: 3.81 cm; Farooq and Dragoi 2019: ~5 cm (inferred from Figure 1J); Stella et al. 2019: ~9 cm (Figure S2); Farooq et al. 2019: 7.6 cm; Grossmark and Buzsaki 2016: 5.14 cm; Ólafsdóttir et al. 2017: 17.5 cm). We also note that we illustrate the application of the state space algorithm to data from running in Figure 2B of Gillespie et al. *Biorxiv* (2021), which clearly illustrates that it can capture the dynamics of theta sequences.

We also do not think that our model is biased to over-representing stationary dynamics because we observed similar distributions of speeds when using the MAP estimate (which picks the most likely position without the state space modeling component) with 20 ms time bins (Figure 6B).

10. What happens when a 20 ms time bin is used for state space modelling? Does it reduce the proportion of spatially-coherent events?

We do not have reason to believe that larger time bins would reduce the proportion of spatially coherent events because our analysis using the MAP estimate and 20 ms bins (Figure 6B) rather than the state space model (Figure 6D) also showed many slow events. In any case, we emphasize that there are serious issues with using longer bins for understanding the structure of the data because it introduces arbitrary boundaries between sets of spikes. Some papers try to overcome this with a sliding window, which partially smooths over those boundaries, but even there, the data are chunked into bins that are larger than necessary (e.g. 10 ms for a 20 ms sliding window that is offset by 10 ms each step) and we argue that there is no reason to introduce these arbitrary boundaries.

11. Figure 5. When whole ripples are classified as stationary, stationary-continuous mixture and so on, how is this classification done for the whole ripple? The classification of dynamics is done for each 2 ms bin, but what criteria are used to classify the whole ripple? Please provide more details on the intermediate steps. For instance, if an event is stationary, but only has one 2 ms bin which deviates, is it considered stationary-continuous?

Any time bin that meets the criteria is included. However, the majority of our dynamics have a duration much longer than a single time bin (Figure 5B). We have further clarified this in the methods (page 26).